



# The need for carbon emissions-driven climate projections in CMIP7

Benjamin M. Sanderson[1], Ben B.B. Booth[2], John Dunne[3], Veronika Eyring[4,5,] Rosie A. Fisher[1], Pierre Friedlingstein[6], Matthew J. Gidden[7, 20], Tomohiro Hajima[8], Chris D. Jones[2,15], Colin Jones[14], Andrew King[9], Charles D. Koven[10], David M. Lawrence[11], Jason Lowe[2], Nadine Mengis[12], Glen P. Peters[1], Joeri Rogelj[7,13], Chris Smith[2,14], Abigail C. Snyder[17], Isla R. Simpson[11], Abigail L.S. Swann[16], Claudia Tebaldi[17], Tatiana Ilyina[18], Carl-Friedrich Schleussner[19,20], Roland Séférian[21], Bjørn H. Samset[1], Detlef van Vuuren[22], Sönke Zaehle[23]

[1] CICERO, Centre for International Climate and Environmental Research, Oslo, Norway
[2] Met Office Hadley Centre, Exeter, United Kingdom
[3] NOAA/OAR Geophysical Fluid Dynamics Laboratory, Princeton, United States
[4] Deutsches Zentrum für Luft- und Raumfahrt e.V. (DLR), Institut für Physik der Atmosphäre, Oberpfaffenhofen, Germany.
[5] University of Bremen, Institute of Environmental Physics, Bremen, Germany.
[6] University of Exeter, United Kingdom
[7] IIASA, Laxenburg, Austria
[8] Japan Agency for Marine-Earth Science Technology, Yokohama, Japan
[9] School of Geography, Earth and Atmospheric Sciences, University of Melbourne,  Australia
[10] Lawrence Berkeley National Laboratory, Berkeley CA, USA
[11] National Center for Atmospheric Research, Boulder, CO USA
[12] GEOMAR, Helmholtz Centre for Ocean Research Kiel, Kiel, Germany
[13] Grantham Institute - Climate Change and Environment, Centre for Environmental Policy, Imperial College London, London, United Kingdom
[14] University of Leeds, United Kingdom
[15] School of Geographical Sciences, University of Bristol, UK
[16] University of Washington, Seattle WA, USA
[17] Joint Global Change Research Institute (JGCRI), College Park MD, USA
[18] Universität Hamburg, Helmholtz-Zentrum hereon, Max Planck Institute for Meteorology, Hamburg, Germany
[19] Humboldt University, Berlin, Germany
[20] Climate Analytics, Berlin, Germany
[21] Centre National de Recherches Météorologiques, Toulouse, France
[22] PBL Netherlands Environmental Assessment Agency, Netherlands
[23] Max Planck Institute for Biogeochemistry, Jena, Germany

*Correspondence to*: Benjamin M. Sanderson (benjamin.sanderson@cicero.oslo.no)

**Abstract.** Previous phases of the Coupled Model Intercomparison Project (CMIP) have primarily focused on simulations driven by atmospheric concentrations of greenhouse gases (GHGs), both for idealized model experiments, and for climate projections of different emissions scenarios.  We argue that although this approach was pragmatic to allow parallel development of Earth System Model simulations and detailed socioeconomic futures, carbon cycle uncertainty as represented by diverse, process-resolving Earth System Models (ESMs) is not manifested in the scenario outcomes, thus omitting a dominant source of uncertainty in meeting the Paris Agreement.  Mitigation policy is defined in terms of human activity (including emissions), with strategies varying in their timing of net-zero emissions, the balance of mitigation effort between short-lived and long-lived climate forcers, their reliance on land use strategy and the extent and timing of carbon removals. To



explore the response to these drivers, ESMs need to explicitly represent complete cycles of major GHGs, including natural processes and anthropogenic influences. Carbon removal and sequestration strategies, which rely on proposed human management of natural systems, are currently represented upstream of ESMs in an idealized fashion during scenario
development. However, proper accounting of the coupled system impacts of and feedback on such interventions requires explicit process representation in ESMs to build self-consistent physical representations of their potential effectiveness and risks under climate change. We propose that CMIP7 efforts prioritize simulations driven by CO2 emissions from fossil fuel use, projected deployment of carbon dioxide removal technologies, as well as land use and management, using the process resolution allowed by state-of-the-art ESMs to resolve carbon-climate feedbacks. Post-CMIP7 ambitions should aim to
incorporate modeling of non-CO2 GHGs (in particular sources and sinks of methane) and process-based representation of carbon removal options. Such experiments would allow resources to be allocated to policy-relevant climate projections and better real-time information related to the detectability and verification of emissions reductions and their relationship to expected near-term climate impacts. Such efforts will provide information on the range of possible future climate states including Earth system processes and feedbacks which are increasingly well-represented in ESMs, thus forming a critical and
complementary pillar underpinning proposed km-scale climate modeling activities and calls to better utilize novel machine learning approaches.

## 1 Introduction

Past phases of the Coupled Model Intercomparison Project (CMIP)(Meehl et al. 2007; Taylor, Stouffer, and Meehl 2012; Eyring et al. 2016) have been the principal source of process-based climate and Earth system modeling outcomes for
IPCC Assessment Reports ("Climate Change 2021: The Physical Science Basis" n.d.). The vast majority of CMIP experiments have considered boundary conditions where concentrations of greenhouse gases are prescribed, both in the implementation of idealized simulations and in future scenarios which inform climate policy (O'Neill et al. 2016; Arnell et al. 2004; van Vuuren et al. 2011; Gillett et al. 2016).

In the two most recent IPCC cycles, scenario experiments have been defined in terms of Representative Concentration
Pathways (Moss et al. 2010), which define futures in terms of approximate end-of-century radiative forcing levels to provide a set of consistent scenarios to be used in climate research, and to provide multiple model-informed climate impact assessments at different warming levels. In CMIP6, scenarios were jointly defined in terms of a 2 dimensional SSP-RCP space (SSPs (Riahi et al. 2017)), where RCPs (Representative Forcing Pathways) explored a wide range of global mean end-of-century radiative forcing targets and Shared Socioeconomic Pathways (SSPs) sampled sociological challenges to mitigation  and
adaptation (O'Neill et al. 2016; Riahi et al. 2017).

The SSP design is concentration-driven, with scenarios defined by their climate response. For example, SSP1-2.6 is a scenario which is designed to achieve a radiative forcing of 2.6 Wm$^{-2}$ in 2100. This is achieved by linking the Integrated Assessment Model (IAM) with a simple climate model (SCM), to solve for a desired climate outcome (Riahi et al. 2017). To meet the





predefined climate target, the IAM-SCM integration is iteratively solved with either carbon emissions constraints or carbon
price trajectories until the climate target is met with sufficient accuracy ("GCAM Documentation" n.d.; "IMAGE Documentation" n.d.; "MESSAGE Documentation" n.d.). For the SSP design, all IAMs used the same simple climate model (MAGICC6.8) to ensure they reached the same forcing level in 2100 (Riahi et al 2017). In the CMIP pipeline, the resulting emissions from each IAM SSP scenario are harmonized to a common historical dataset, any missing emissions infilled, and then multi-gas concentration pathways are estimated by a common SCM (Meinshausen et al. 2020), to be used as boundary
conditions for ESM simulations in future scenario projections, together with pre-computed spatial information on land use and aerosol emissions (Feng et al. 2020; Hurtt et al. 2020).

The RCP framework was developed to allow computationally expensive ESM simulations to start computing simulations, while in parallel, Integrated Assessment Models (IAMs) were used to develop scenarios consistent with the RCPs. This step was felt necessary to ensure results were delivered in time for the IPCC Fifth Assessment Report (Moss et al., 2010). As such,
SSP-RCPs use concentrations as a definitional anchor point. In this framework, Earth System uncertainties as a function of concentrations are estimated by climate models (in practice, by the CMIP ensemble, Figure 1). This has pragmatic advantages in terms of coordinating research across climate disciplines, but excludes carbon cycle uncertainties. The concentration-based framework has no structurally consistent mechanism for representing these uncertainties in a process-resolving fashion - the IPCC AR6 WG1 report relied on emulators which were informed indirectly by CMIP models, but climate and carbon
uncertainties were independently calibrated("IPCC AR6 Working Group 1: Technical Summary" n.d.).

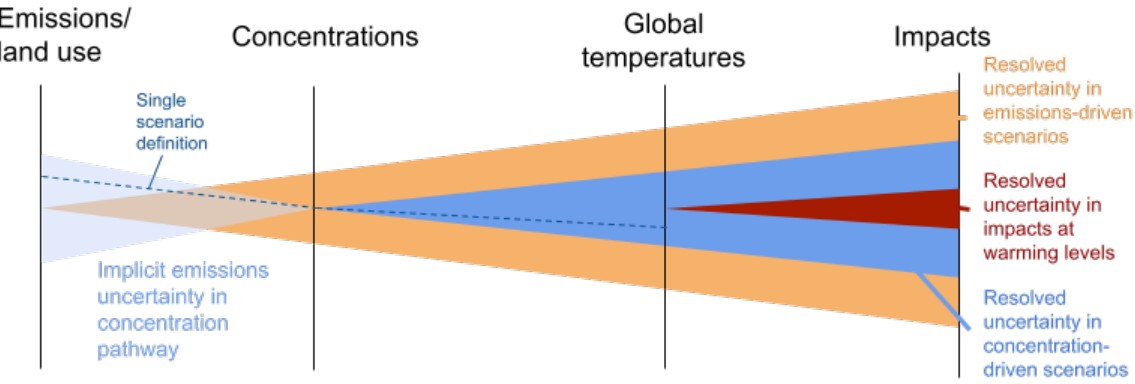

**Figure 1: A conceptual illustration of the propagation of uncertainty using concentration and emissions-based anchor points**

**1.1 The trouble with compatible emissions**

To date, CMIP phases have primarily represented anthropogenic emissions as a residual in concentration-driven simulations (P. Friedlingstein et al. 2006; Chris D. Jones et al. 2016), thereby computing 'compatible emissions' consistent with the prescribed concentrations. This is achieved by assessing the residual flux of carbon which would be necessary to balance the



internal carbon budget of an ESM simulation which is run in concentration-driven mode. However, in idealized simulations,
these compatible emissions pathways often exhibit sharp discontinuities which render them less relevant to real-world carbon-climate dynamics (Koven, Sanderson, and Swann 2023), and in scenarios there are often significant differences between the carbon cycle representations in the original IAM structure and the ESM, such that the compatible emissions are conceptually distinct from the original scenario design (Koven et al. 2022) (and Figure 2). For ambitious mitigation scenarios such as SSP1-RCP2.6, these differences account for a significant variation in the total cumulative emissions consistent with the prescribed
concentration pathway. As the scenario literature increasingly focuses on mitigation strategy relevant to the Paris agreement (Rogelj, Huppmann, et al. 2019; Sognnaes et al. 2021), it becomes increasingly necessary for ESM simulations to accurately represent both historical emissions and the outcomes of emissions scenarios which are consistent with the socioeconomic trajectories they are meant to represent.

A second issue with compatible emissions is the model-dependent ambiguity in their computation. Because compatible
emissions are computed as a residual, after accounting for carbon in the land surface, ocean and atmosphere, it is necessary that all models output the needed fields to account for the complete carbon budget. However, not all models output biome integrated carbon fluxes - requiring their reconstruction from constituent components. More fundamentally, there is inconsistency in the carbon pools and land use processes represented in different models - confusing the interpretation of the compatible emissions (Liddicoat et al. 2021). Furthermore, compatible emissions can only diagnose the fossil-fuel component
(C. Jones et al. 2013). This meant for example that IPCC AR6 had to mix ESM output for diagnosed fossil fuel emissions and IAM-based scenario data on land-use emissions in creating synthesis figures such as WG1-SPM.7.

In addition, ESMs calculate land use, land use change and forestry (LULUCF) emissions dynamically based on the changing land-use patterns which can markedly differ from the original LULUCF fluxes computed in IAMs (Quesada et al. 2018; Wilkenskjeld et al. 2014), and these differences are counterintuitively manifested in the compatible emissions which, in theory,
should represent fossil fuel emissions. This also means that compatible emissions calculated in SCMs are not comparable with ESM estimates, because aggregate LULUCF emissions are exogenously prescribed in most SCMs - creating discrepancies between SCM and ESM estimates of remaining carbon budgets for given warming levels (Millar et al. 2017).





**Figure 2: Compatible emissions for a range of scenarios and Earth System Models in CMIP6, showing the IAM internally calculated CO2 emissions (dotted black, combined fossil and land use), and the compatible emissions in CMIP6 ScenarioMIP simulations (colored lines), derived from concentration driven simulations but with a diagnostic carbon cycle that represents sources and sinks in the land and ocean and calculating residual emissions that would balance the model's carbon budget. Historical total anthropogenic emissions** (P. Friedlingstein et al. 2022) **are shown for context.**

Assessing compatible emissions for CMIP6 scenarios underlines that there are significant differences in the simulated compatible emissions amongst ESMs (Figure 2). For example, in the concentration driven SSP1-2.6 scenario in CMIP6, ESM-



simulated net-zero dates measured in terms of compatible fossil fuel emissions ranged from 2076-2086, compared with the IAM estimate of 2076 (Gidden et al. 2019; van Vuuren et al. 2017) (Figure 2), resulting from differences between the IAM and ESM representations of carbon-climate dynamics.

The only emissions-driven scenarios in CMIP6 took place as part of C4MIP (Chris D. Jones et al. 2016) , repeating high
emissions scenarios (*esm-SSP5-RCP8.5*) and an extreme overshoot scenario (*esm-SSP5-RCP3.4-over*) with a small subset of models. Notably, these scenarios were chosen to inform assessments of carbon feedbacks under high emissions (but they are not themselves considered to represent realistic near-term futures (Hausfather and Peters 2020b)). As a result, multi-model ESM results from the CMIP6 scenario effort as presented in IPCC-AR6-WG1 (e.g. AR6-WG1-Fig4.11) exclude an assessment of carbon cycle uncertainty (Tebaldi et al. 2021; Lee et al., n.d.). Where carbon-climate feedbacks were considered in IPCC
consideration of SSP projections (e.g. AR6-WG1-Fig4.35), this was achieved by probabilistic SCM ensembles informed by idealized ESM experiments to inform carbon feedback parameter uncertainty (Chris D. Jones et al. 2016; P. Friedlingstein et al. 2006; "IPCC AR6 Working Group 1: Technical Summary" n.d.).

In this perspective, we argue that the increasing sophistication and stability of emissions-driven model configurations relevant for modelling greenhouse gas cycles means that this approach can now be reassessed. The urgent need for process-based
information on the mitigation effectiveness of fossil fuel emission reductions, carbon dioxide removal, and land use policies, requires a framework for the increased inclusion of emissions-driven experiments in upcoming CMIP cycles, in the presence of heterogeneous model complexity, timeline constraints and technological challenges.

## 2    The need for emissions-driven ESM scenarios

Climate policy is framed in terms of emissions - naturally focussing on the elements that can inform mitigation decisions, such
as emission benchmarks, carbon budgets and the timing of net-zero. In addition, emissions-driven climate metrics (Arora et al. 2020) such as the transient climate response to cumulative emissions of carbon dioxide (TCRE, (Chris D. Jones and Friedlingstein 2020) and the Zero Emissions Commitment (ZEC, (Chris D. Jones et al. 2019) are important and policy-relevant summary quantifications of the Earth System response to climate mitigation efforts. As of today, countries have committed to achieving climate targets, including net-zero targets, under the Paris Agreement, that constrain the future emissions space.
Consistency of simulations with policy constraints is key to providing policy relevant information.

However, the dominance of concentration-driven scenarios means that CMIP6 does not contain self-consistent simulations of mitigation strategy and their climate outcome in Earth System Models. As a result, though IAM simulations already frame scenarios in terms of emissions pathways (Sognnaes et al. 2021), the simplified internal representation of climate and carbon processes does not allow for a comprehensive assessment of the underlying carbon cycle uncertainties associated with the
scenario tradeoffs, generally relying on simple climate models to represent uncertainty in carbon-climate feedbacks (Nauels et al. 2017; Bodman, Rayner, and Jones 2016; Damon Matthews et al. 2021; Duncan Watson-Parris and Smith 2022), where idealized ESM results may be indirectly used in the calibration of the simple climate model parameter distributions. However,





while this approach is convenient, bypassing process-resolving carbon cycle uncertainty in the assessment of mitigation strategy outcome has risks. A CMIP ensemble with a primary focus on emissions-driven scenarios, starting with $CO_2$ emissions in CMIP7 but with a longer term objective to represent human activity through diverse emissions or land management, would allow ESM scenarios to represent real-world climate policy and its outcomes.

## 2.1 Key science needs for emissions-driven models

This emission-driven CMIP7 strategy would enable four key scientific benefits, which we outline in this section: 1) process-resolved assessment of carbon removal assumptions which underpin the capacity for climate temperature overshoot, 2) trade-offs between fossil fuel emissions, carbon removals, land use change, and short lived climate forcers on regional scales including relevant feedbacks, 3) integrated process-resolution of system thresholds, nonlinearities, and risks which might exacerbate climate impacts and modify Earth System feedbacks in warmer climates and 4) relevant simulations to inform the verification of mitigation activity.

### 2.1.1 Qualifying assumptions on carbon removal and overshoot

The plausibility and effectiveness of the gigatonne-scale carbon dioxide removal implied by mid- to high-mitigation scenarios is a key uncertainty (Marcucci et al. 2019) for end-of-century warming outcomes, given that the majority of the world's economy has pledged net-zero CO2 or GHG targets which are themselves conditional on significant amounts of carbon dioxide removal (Grant et al. 2021). Increasingly, this assumed feasibility of net global removal of carbon extends to climate overshoot pathways, where the temperature limits of the Paris Agreement are temporarily exceeded. High level communication of climate science often frames the possibility of a temperature overshoot as a given; for example headline statement B.7 of the IPCC AR6 synthesis report presents the option of temperature overshoot in certain terms: "If warming exceeds a specified level such as 1.5°C, it could gradually be reduced again by achieving and sustaining net negative global CO2 emissions." .

The plausibility of large scale CDR is subject to uncertainties which are not captured in the current IAM and ESM modeling framework. Interventions will cause biophysical and biogeochemical feedbacks on the climate system that are not currently represented by the IAM-simple climate models used to define scenarios (Koch, Brierley, and Lewis 2021; Luyssaert et al. 2018; Melnikova et al. 2023).

For land-based CDR approaches, the carbon sinks assumed within IAMs for a given land use transition are themselves subject to climate-induced risks due to warming (drought, wildfire, insect outbreaks (Anderegg et al. 2022; Nathan G. McDowell and Allen 2015; Nate G. McDowell et al. 2020) which are not taken into account in IAM scenarios which rely on approaches such as Bioenergy Carbon Capture and Sequestration (BECCS) for large scale carbon removal (Kato and Yamagata 2014; Muri 2018). In addition, carbon sink strengths themselves respond dynamically to emissions and removals of gases through carbon concentrations, aerosol forcing, and surface ozone (Sonntag et al. 2018; Mengis et al. 2019; O'Sullivan et al. 2021; Zhang et al. 2021) - dynamics which can only be represented in an emission-driven, process resolving model structure.



We can illustrate in Figure 3 the scale of these potential uncertainties in the feasibility of land-based CDR capacity using a
pair of scenarios from CMIP6; the highest emission member of the ScenarioMIP ensemble SSP5-RCP85 and the extreme
overshoot scenario SSP5-3.4-overshoot (Kriegler et al. 2017; Riahi et al. 2017), which assumes a significant amount of BECCS
is deployed in the latter half of the 21st century (with bioenergy crop production of 9PgC/yr  by 2100).  The assumed carbon
removal in the IAM notably exceeds the difference between ESM simulated harvest flux in SSP5-85 (where there is no
deployed BECCS) and SSP5-34-over in all 3 of the models considered (difference between purple and red lines, Fig. 3).  For
one model (CESM2-WACCM), the assumed BECCS flux in SSP5-34-over exceeds the *total* harvest production simulated in
that scenario (including all energy and food-based crops).  Though this scenario pair represents an extreme overshoot, it
illustrates that IAM assumptions about available CDR capacity may at best be inconsistent with ESM simulated fluxes, and at
worst completely unphysical.




**Figure 3: (a) An illustration of total harvest carbon flux as simulated in the SSP5-34-overshoot (solid) and SSP5-85 (dashed) scenarios as simulated by the REMIND-MAGPIE integrated assessment model (black lines), compared with estimates from 3 climate models (colored lines) which completed both simulations. (b) colored lines show the simulated difference in ESMs between harvest carbon flux in SSP5-34-overshoot and SSP5-85. Black lines show bioenergy harvest in the REMIND-MAGPIE IAM for SSP5-34over (solid), SSP5-85 (dashed - negligible in this scenario) and the net carbon removal from BECCS (dotted).**




Ocean carbon balance is also conditional on the climate state and wider mitigation efforts. If net negative emissions are achieved over the land, dissolved carbon will likely be released from the oceans, acting to counter removal efforts (Vichi, Navarra, and Fogli 2013). Ocean based CDR suggestions such as alkalinity enhancement (Fakhraee et al. 2023; Hartmann et
al. 2023) or iron fertilization (Emerson 2019) are also conditional on the wider climate state and can have significant non-local effects on the wider biosphere (Keller, Feng, and Oschlies 2014).

Issues over the feasibility of CDR at scale are compounded by uncertainties in the response of the Earth System to extended periods of net zero or net negative emissions. Much of current understanding stems from highly idealized ESM experiments which have been conducted by only a subset of models (Chris D. Jones et al. 2019; Keller et al. 2018). Such experiments show
that Earth System response to net negative emissions is complex and likely asymmetric, but the lack of extensive process-based ESM simulations of response to net negative emissions leaves significant uncertainties where SCMs and emulators have not been extensively tested or validated. Such uncertainties have bearing on the feasibility of a temperature overshoot, both in terms of the level of mitigation needed to stabilize warming (Jenkins et al. 2022) and the relative timing of net-zero and peak warming (Koven, Sanderson, and Swann 2023).

The current framing mostly neglects the coupled dynamics between carbon removal and the wider climate state, meaning that its utility as a decision-informing exercise is fundamentally limited. As such, concentration-driven mitigation scenarios created through the existing modeling chain may assume land-use and management carbon fluxes from the IAM which are impossible to achieve with the ESM (and perhaps reality) due to ecophysiological limitations of vegetation in a changing climate. An emissions-driven framework would directly assess these risks associated with land-based carbon mitigation (such as through
afforestation, reforestation, forest management, biochar, agricultural soils or BECCS).

An emissions-driven framing is naturally suited to process representation of carbon dioxide removal methods (especially for those methods which rely on the manipulation of natural systems which are to some degree resolved within Earth System Models). Some of these (such as afforestation) are already represented within most ESMs, while others (BECCS, soil carbon enhancement, terrestrial and marine alkalinity enhancement, blue carbon enhancement) are represented to a lesser degree or
not at all. A dedicated activity within CDRMIP could assess the effectiveness of different approaches in a semi-idealized context under different climate background states. Such an activity could aid in the interpretation of emissions-driven scenario simulations in CMIP7 and provide a pathway to the inclusion of a wider range of CDR technologies in CMIP8 and beyond.

### 2.1.2 The need for activity-driven scenarios

The 'illustrative pathways' provided by the IPCC special report on 1.5 Degrees of Global Warming (V. Masson-Delmotte et
al. n.d.) and in the 6th Assessment Report (Wgiii n.d.) span a range of trade-offs between energy system change, economic growth, and carbon dioxide removals. However the SSP-RCP framework used to sample scenarios in CMIP6, by design, assumes that the physical climate space sampled by ESMs can be defined in terms of a single dimension of end of century radiative forcing level.




However, pathways with similar end of century global mean temperature outcomes can be subject to divergent physical
climates and ecological responses. For example, land mitigation strategies have a joint effect on both the carbon cycle, regional
climates (D. Li and Wang 2019) and competition for land use and food security (Qi et al. 2018). The relative level of ambition
and timing of mitigation for different greenhouse gases can change the timing and emergence of climate impacts (Lund et al.
2020) and how (or if) to combine these commitments is a critical challenge for mitigation (Allen et al. 2022).

Human activity changes which impact aerosol emissions can have a marked effect on regional patterns of climate change (Liu
et al. 2018), and shifts in centers of aerosol emissions can have significant impacts on observed rates of warming and regional
precipitation changes which can compound and modify the background greenhouse gas effects (Bjørn H. Samset et al. 2019).
Aerosol processes can also be intricately linked with carbon uptake (O'Sullivan et al. 2021; Zhang et al. 2021), impacting both
the interpretation of past carbon cycle evolution and future carbon uptake in areas with large aerosol concentrations/surface
ozone (e.g. S. Asia/Africa).

These dimensions increasingly dominate many of the most pressing questions in climate policy, and process resolving ESMs
are in a unique position to provide self-consistent assessments of climate policies which have both regional, temporal, and
species dimensions. Constructing scenarios which fully explore these dimensions requires scenario definitions which go
beyond end-of-century forcing or temperature level implied in a concentration pathway. Rather, mitigation strategy needs to
be defined in terms of activity and consequence: where human activities include fossil fuel and other industrial emissions,
combined with regionally resolved descriptions of land use change and management.

The hybrid approach proposed in this study considers a set of headline experiments in CMIP7 which are preferentially driven
by carbon and aerosol emissions, with prescribed values for other atmospheric components. Such an approach would be
supported by continued activities in RFMIP (Pincus, Forster, and Stevens 2016) to provide diagnostics of global aerosol
emissions-forcing-feedback dynamics, but also in AerChemMIP (Collins et al. 2017) which in CMIP6 assessed the role of
aerosol forcing process uncertainty in future simulations. Finally, there are some activities which did not exist under the
CMIP6 platform which could be highly valuable in the increased understanding of emissions-driven processes. A dedicated
activity to assess the role of regional aerosol emissions in this uncertainty (Wilcox et al. 2022) would address the growing
consensus that shifts in regional emissions intensity has a large and detectable climatic impact (Bjørn H. Samset et al. 2019).
And, for those models capable, dedicated activities to assess the coupled dynamical response of the Earth System to non-$CO_2$
gases such as $N_2O$ and $CH_4$ would provide critical groundwork for their eventual representation in following CMIP activities.

### 2.1.3    Resolving compound tipping points and adaptation challenges as a function of emissions

The potential for nonlinearities and tipping points in the climate system is frequently raised as a motivator for urgent emissions
cuts (Lenton et al. 2019), and often framed in terms of temperature thresholds (for example, in discussion of whether rapid and
irreversible changes might be triggered if 1.5C of warming above pre-industrial levels is exceeded (Armstrong McKay et al.
2022)) - but introducing previously ignored nonlinearities can complicate how thresholds defined in terms of temperature map
onto mitigation risks. Some of these previously discussed system thresholds have the potential to alter global scale carbon-



climate feedbacks and dynamics e.g. the risk of crossing cryosphere thresholds (Kloenne et al. 2022), forests may be subject to dieback or changes in carbon sink efficacy (Chai et al. 2021) and increased stratification of the ocean may change its heat and carbon uptake dynamics (Bourgeois et al. 2022).

As such, tipping points and emissions are intricately tied together and Earth System Models are natural tools for simulating how they might interact, with increasingly complete and sophisticated process resolution for ecosystem and cryosphere and ocean processes. Understanding how these nonlinearities combine, and relate to a wider mitigation strategy requires the processes to be simulated in a self-consistent framework in the context of a emissions-driven mitigation scenario where carbon-climate feedbacks are interactively resolved.

This argument extends to adaptation planning, where ESM results from concentration-driven simulations are often currently framed in terms of expected impacts at given warming levels (Jevrejeva et al. 2018; Lwasa et al. 2018; Valérie Masson-Delmotte et al. 2022; Travis, Smith, and Yohe 2018) rather than impacts under given emissions pathways (Drouet et al. 2021; Wiebe et al. 2015). As such, adaptation planners have no simple means of assessing the range of plausible hazards consistent with a given level of climate policy. Emissions-driven simulations could help fill this gap, while still allowing impacts to be

framed in terms of warming levels as they are with existing ensembles.

### 2.1.4    Verification of emissions reductions

The 2028 Global Stocktake will be the next major global assessment of progress towards Paris Agreement goals. This requires increasing understanding of how to quantify and verify national emissions reductions. Existing approaches for the detection and attribution of observed climate changes to different historical anthropogenic activities rely predominantly on models in

concentration driven mode (Hegerl and Zwiers 2011). However, with increasing focus on mitigation activity and the verification of reductions in terms of climatic variables (such as greenhouse gas concentrations, temperatures or heat uptake)(Peters et al. 2017), it makes sense to consider the detection problem in terms of emissions - when can the benefits of mitigation activity be observed?

As climate mitigation ambition ramps up, there is a growing expectation that emissions will change their recent historical

trend, initially with slower growth, then a peak, followed by a decline. Already, global $CO_2$ emissions have slowed from 3% per year growth in the 2000s to 1% per year growth in the 2010s (Pierre Friedlingstein et al. 2022). An increasingly relevant question will then be to what degree any reductions will be detectable in terms of observed climate variables and near-term warming (McKenna et al. 2020; B. H. Samset et al. 2022) and, potentially, climate impacts themselves (Mendez and Farazmand 2021; Ciavarella, Stott, and Lowe 2017). These questions are of relevance for the justification of climate policy,

both globally and at the country level, and for planning for potential near-term impacts and for assessments of liability for climate damages.

Modeling to support such activity requires a joint assessment of land, ocean and atmospheric carbon pool and human activity in a self-consistent framework (T. Ilyina et al. 2021). Land sinks are of particular relevance in the context of the Global Stocktake process which assesses national-level progress in the context of meeting obligations under the Paris Agreement. In



this process, many countries offset a fraction of their emissions using managed land within their borders which is currently assessed to act as a carbon sink (Grassi et al. 2021). Understanding the robustness of these sinks in present and future divergent climates is thus critical in assessing the degree to which countries can rely on such sinks to substitute for emissions reductions on different timescales (Giebink et al. 2022).

In the atmosphere, efforts to detect emissions reduction from globally averaged atmospheric concentrations have not yet
succeeded. It was expected that a two percentage point change in the growth rate of $CO_2$ emissions could be detected in the atmosphere with reasonable confidence after about 10 years (Peters et al. 2017). A possible explanation for the lack of signal is our inability to fully model and explain the inter-annual variability in climate-carbon feedbacks, which could be offsetting a part of the expected change in trend (Spring, Ilyina, and Marotzke 2020). In the years ahead, when emissions are hopefully declining, there will be a need to understand how the carbon cycle may respond with carbon-climate feedbacks potentially
offsetting some of the expected declines in the atmospheric growth rate. Such experiments have to date been idealized (Keller et al. 2018; Chris D. Jones et al. 2019), but there remains a need for integrated simulation to explore the interaction of natural carbon feedbacks with process-resolving CDR and non-CO2 emission pathways.

To date, attempts to verify emissions reductions as a function of atmospheric concentrations have been conducted in simple climate models (Abdulla et al. 2023), by adjustments computed from compatible emissions in Earth System Models (Spring,
Ilyina, and Marotzke 2020) or by using atmospheric inversion models to compute emissions consistent with prescribed concentrations (Deng et al. 2022). Each of these is a pragmatic approach to verifying emissions reductions, but none provide a fully self-consistent internally generated representation of the chain of causality from emissions to concentrations.

Such questions could be addressed in DAMIP (Gillett et al. 2016)) or other activities using a combination of idealized and realistic simulations: (1) idealized experiments where $CO_2$ emissions reduce at a fixed rate to detect timing of signal emergence,
(2) emissions-driven single forcer experiments to assess the detectability and linearity of the historical climate response to different anthropogenic emissions. As such, emissions-driven simulations would provide a critical complement to existing verification efforts, potentially including counterfactual scenarios which could illustrate when mitigation policy implementation becomes detectable in terms of atmospheric concentrations or climate impacts (Tebaldi and Friedlingstein 2013).

**3    The coming of age of emissions-driven Earth System Models**

Past CMIP phases designed experiments to exploit the existing modeling capacity in major Earth System modeling centers at the time of experimental design, motivated by dominant uncertainties and pilot studies in the literature (Meehl et al. 2007; Taylor, Stouffer, and Meehl 2012; Eyring et al. 2016). Early climate simulations used atmospheric-only models to diagnose radiative feedbacks (Cess et al. 1989). CMIP2 era coupled experiments generally exploited radiative
flux corrections to maintain a stable ocean temperature (Covey et al. 2003), and a parallel Atmospheric Model Intercomparison Project (AMIP) process remained to understand atmospheric feedbacks without the added complexities





of ocean coupling (Lawrence Gates et al. 1999). The presence of an intercomparison project fostered rapid improvements in coupled simulation such that by the time of the CMIP3 ensemble (Meehl et al. 2007), there was increasing acceptance that resolving coupled ocean-atmosphere processes was key to understanding climate

projections (Frame et al. 2006), and models were rapidly advanced so that they could maintain stable climates without flux corrections.

Over the last 20 years, the scope of process resolution in climate models has further expanded (Figure 3), and the increasing complexity of both atmospheric chemistry and aerosol treatment has increased the degree to which some emissions are already represented in many climate models and interact with climate feedbacks (Thornhill et al. 2021). The evolution of aerosol

treatment from CMIP3 to CMIP5 to CMIP6 has seen a non-uniform tendency for models to represent aerosol indirect effects on clouds, and emissions-driven aerosol processes (interactive treatment of aerosols have been included in some fraction of Earth System Models since CMIP5 (Eyring et al. 2016), and stratospheric aerosols have been included since CMIP3 (Meehl et al. 2007)). CMIP6, in particular (Eyring et al. 2016) introduced an tiered experimental design which accommodated models with varying levels of aerosol and atmospheric chemistry implementation in scenario experiments, supported by dedicated

sub-MIPs to assess processes (in AerChemMIP) and effects of different forcers (in RFMIP).



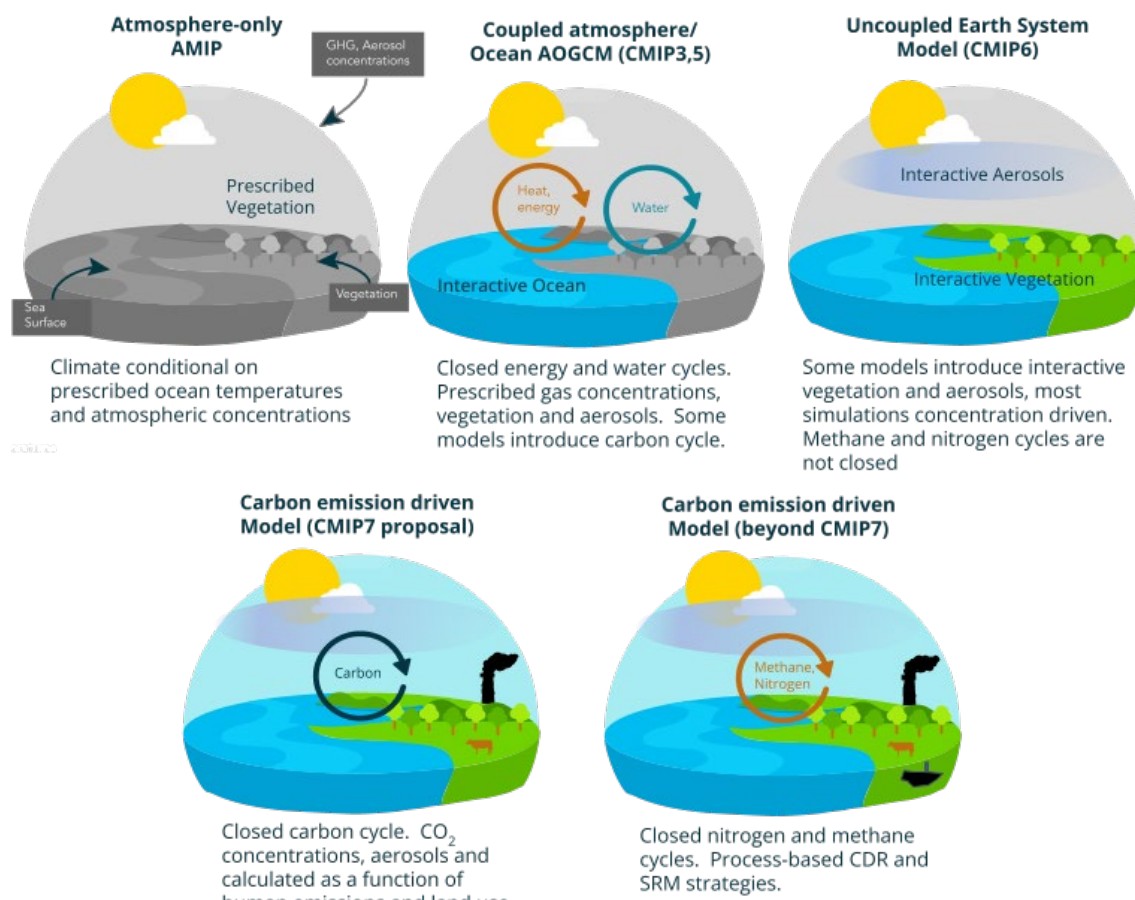

**Figure 4: the evolving dominant paradigm in different generations of CMIP, including this study's recommendations for CMIP7 and CMIP8**

### 3.1 Default coupled carbon cycle modeling in CMIP

Past phases of CMIP have defaulted to concentration-driven scenarios, but models capable of running with a closed and interactive carbon cycle have been developed by some centers for over two decades (Cox et al. 2000), with intercomparison efforts for coupled carbon Earth System Models coming soon after (P. Friedlingstein et al. 2006; C. D. Jones 2020). However, despite increasing acknowledgment of the central role of coupled climate-carbon dynamics in determining the outcome of mitigation policies (Allen et al. 2009; Holden et al. 2018), only 19 out of 82 CMIP6 model configurations participated in the

Coupled Climate–Carbon Cycle Model Intercomparison Project (C4MIP) in CMIP6 ("Cmip6 Data Search" n.d.), though these models vary in resolved processes (12 resolving carbon-climate interactive feedbacks, 5 resolving phytoplankton biophysical interactions, 3 resolving biogenic aerosol-cloud feedbacks and no models representing non-$CO_2$ biogeochemical cycle feedbacks (Séférian et al. 2020)).



The operational computational cost of modeling Earth System Processes is a factor in development priorities, but is not
prohibitive. The most notable increase in expense (relative to physics-only simulations) in simulating the carbon cycle arises
due to the number of tracers required in the biogeochemical models (Kwiatkowski et al. 2014). As such, an ESM
configuration requires some tradeoffs between horizontal and vertical resolution, number of tracers and the complexity of
chemistry and aerosol representation- with the potential for multiple configurations with comparable computational costs with
focus on Earth System processes or resolution respectively (Dunne et al. 2020). However, because for most CMIP-class
models, the atmospheric component is significantly more expensive (Danabasoglu et al. 2020; Dunne et al. 2020; Hedemann,
Hohenegger, and Ilyina, n.d.), land and ocean biogeochemistry (BGC) can be run in parallel with the atmosphere - somewhat
increasing the CPU requirements, but not the overall run-time of the simulation on parallel High Performance Computing
(HPC) systems.

Recent ESM development efforts have shown that spinning up oceanic carbon cycles can be achieved on the same timescale
as for deep ocean heat content, which is necessary for any atmosphere-ocean coupled configuration (Lindsay et al. 2014; Yool
et al. 2020)) (although the exact details of how spinup is achieved can impact residual trends (Séférian et al. 2016)). Moreover,
there are a number of promising efforts to accelerate the spinup of the physical ocean (Lindsay 2017; Singh et al. 2022) and
land (Lu et al. 2020; Sun et al. 2023), further lowering the technical barriers to contributing with stable interactive carbon
configurations. Other efforts have improved the parallelisation of BGC tracers (Linardakis et al. 2022) and grid coarsening
(Berthet et al. 2019)- allowing for the better exploitation of HPC infrastructure to run more comprehensive carbon resolving
simulations without increases in wallclock time.



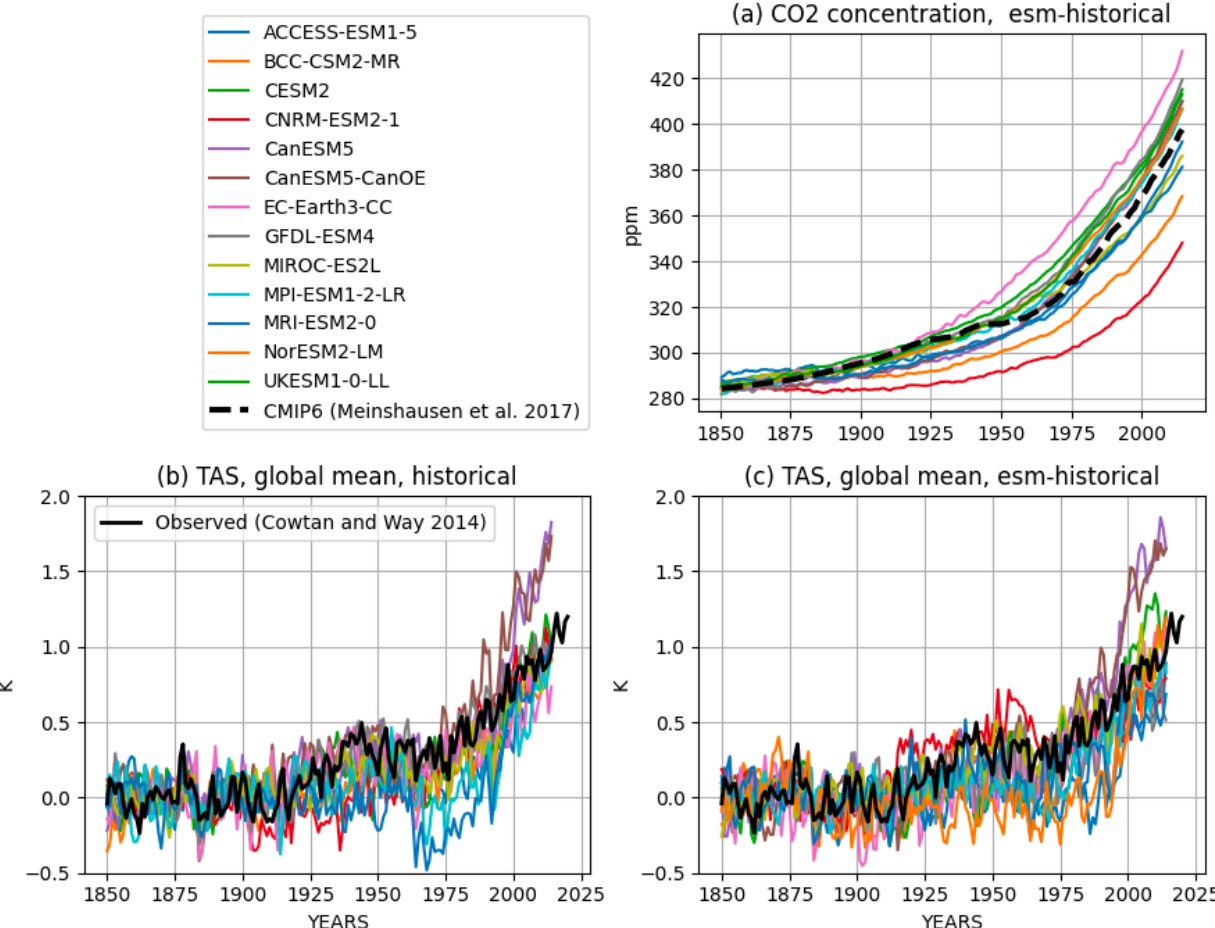

**Figure 5: Carbon dioxide concentrations (a) and temperature anomalies (c) in emissions-driven historical simulations in CMIP6, and temperature anomalies concentration-driven historical simulations (b).** Temperature anomalies (Cowtan and Way 2014) **are calculated from the 1850-1900 average.**

Another perceived challenge is the additional degrees of freedom associated with calibrating the coupled climate-carbon cycle system to reproduce both the joint evolution in historical concentrations of climate forcers and the historical warming increases. CMIP6 esm-historical simulations show most models (10 out of 13 models in C4MIP) fall within a range of CO2 concentration range of 40ppm. Although this is significantly greater than the observational uncertainty (about 0.1ppm(Pierre Friedlingstein et al. 2022; Lee et al., n.d.)), it is not a significant factor in the model uncertainty in warming represented by the distribution of historical warming in CMIP6  simulations and their concentration-driven historical analogs for models which completed both experiments (Figure 5b,c) - indicating that $CO_2$ biases from prognostic carbon are not a major source of error in reproducing historical warming trends compared with other factors such as climate sensitivity or historical forcing which dominated biases in CMIP6 (Papalexiou et al. 2020).



Nonetheless, emissions-driven experiments in the 'central' DECK/Historical part of CMIP6 were limited to *esm-historical* and *esm-picontrol* (Eyring et al. 2016). Further, the DECK required independent *picontrol* and *esm-picontrol* simulations from an ESM, and highlighted the importance of large ensemble sampling for the historical simulation. In practice, for models

which conducted the ESM historical simulation *esm-hist*, it was generally without initial condition sampling - presenting an obstacle for the assessment of the role of internal variability in carbon cycle feedbacks, and for signal emergence of coupled Earth System processes (H. Li and Ilyina 2018) and near-term initialized climate prediction systems (H. Li et al. 2023) which enable near-term prediction of atmospheric $CO_2$ concentrations, air–sea and air–land carbon fluxes.

The limited ESM-DECK experiments in CMIP6 were supported by process understanding from idealized carbon cycle

feedback experiments, including the globally aggregated effects of idealized carbon dioxide removal in CDRMIP (Keller et al. 2018), metrics of carbon cycle feedbacks in C4MIP (Chris D. Jones et al. 2016) and ZECMIP (Chris D. Jones et al. 2019) and the physical and carbon effects of land use change in LUMIP (Lawrence et al. 2016) and LS3MIP (van den Hurk et al. 2016). Although C4MIP included some emissions-driven scenarios - (esm-*ssp585* and *esm-ssp534-over*), these represent very large near-term emissions which are distant from contemporary policy discussions (Hausfather and Peters 2020a).

As such, we argue that in order to provide robust information for both adaptation and mitigation, it is equally important to sample inter-model uncertainties in future atmospheric CO2 concentrations as for the physical climate response to a single trajectory of CO2. This requires a change in prioritization in the DECK, ScenarioMIP, and elsewhere in CMIP, with default control, historical, and projection simulations run in emissions-driven configuration, with concentration driven options used as a fallback for models which cannot process emissions. Such a reprioritization would enable modeling centers to more

efficiently use resources to focus on Earth System uncertainties (including physical and carbon cycle elements), rather than splitting resources.

## 3.2 The need for a post-CMIP6 coordinated effort on activity-driven carbon cycle modeling

The status quo which defined the default configurations in CMIP6 and earlier phases is now changing. Models can increasingly resolve vegetation and soil carbon dynamics including permafrost, as well as marine biogeochemical cycles. For many ESMs,

the capability to represent these processes now exists, but relatively little work has been done thus far to comprehensively understand how this complexity impacts the trajectory of climate, especially under deep mitigation scenarios, geoengineering proposals, and overshoots.

ESMs can potentially add self-consistent process resolution to a wide range of carbon processes which are currently resolved in scenarios in an *ad hoc* and quasi-empirical fashion. ESMs are already well placed to resolve natural land and ocean carbon

sinks, and are operationally used to quantify these terms today (Pierre Friedlingstein et al. 2022). But in addition to this, they can directly inform the effectiveness and uncertainty associated with land use and management policy, and their coupled interaction with natural sinks (Lawrence et al. 2016). Beyond this, many high ambition scenarios contain significant



requirements for explicit representation of carbon dioxide removal (Fuss et al. 2014; Anderson and Peters 2016) whose plausibility can potentially be assessed when represented in an Earth System Model (Muri 2018).

An emissions-driven scenario framework would allow for the explicit representation of different forms of human activity associated with carbon mitigation, and much of this has already been demonstrated using subsets of ESMs. Carbon removal technologies (such as bioenergy carbon capture and storage) could largely use existing models combined with sub-annual harvest cycles, harvest-age for woody biomass, and a dedicated pool to represent underground carbon storage. Others, such as cultivation and harvesting of oceanic algae (J. Wu, Keller, and Oschlies 2023) or ocean alkalinity enhancement (Keller,

Feng, and Oschlies 2014; Tatiana Ilyina et al. 2013; Burt, Fröb, and Ilyina 2021; González et al. 2018), could be represented with explicit parameterisations (J. Wu, Keller, and Oschlies 2023). And, as discussion of the ethics and risks of solar radiation management intensify (Reynolds 2021; Sovacool 2021), understanding the interaction between geoengineering and ecosystem processes is of paramount importance (Zarnetske et al. 2021) where coupled ESMs are essential in any comprehensive cost-benefit assessment (Sonntag et al. 2018).

Thus, although there is a large and growing body of work assessing mitigation strategy in the context of emission-driven models, much of this to date has been in the context of isolated ESM experiments which do not capture multi-model uncertainty. By adopting a default emissions-driven design, CMIP7 could directly inform the coupled system risks associated with the range of carbon removal and geoengineering strategies which increasingly play an outsized role in the mitigation debate.

## 4 Modeling needs for CMIP7 and beyond

CMIP7 is the next phase of the Coupled Model Intercomparison Project. Though a timeline is not yet officially defined, results would be relevant to inform both the seventh assessment cycle of the IPCC and potentially the Global Stocktake in 2028. This requires the development, calibration and simulation of forcing datasets, Integrated Assessment Model and Earth System Model simulations in time to contribute to reports and assessments informing these international activities.

### 4.1 Tradeoffs and synergies with high resolution climate modeling objectives

Efforts to improve emissions-driven process representation are complementary to other areas of climate model development which have been documented elsewhere and address other key knowledge gaps: the need for kilometer-scale resolution of future climate impacts (Schär et al. 2020) , the quantification of parametric uncertainty (Yamazaki et al. 2021), robust sampling of internal variability (Deser et al. 2020) and making best use of machine learning for computational efficiency and climate

model performance (Harris et al. 2022; Eyring et al. 2021). Some groups have gone further to suggest that climate modeling efforts must pivot to centralized 'digital twins' conducted by a small number of modeling centers to provide global simulations at kilometer scale resolution (Bauer, Stevens, and Hazeleger 2021).





However, we argue here that it is impossible to address all of the current knowledge gaps with a single modeling strategy focused on delivering results at a convection-permitting resolution. To illustrate, Figure 6 shows the approximate current

tradeoff between horizontal resolution and throughput on current high performance computational infrastructure. For the operational CMIP6 resolution in the range of 50-100 km grid size, models on current representative operational modeling center hardware achieve between 1-100 simulated years per day. This allows the production of ensembles of century scale simulations on a timeline of weeks to months. Current highest resolution 3 km 'convection permitting' models achieve 1-10 simulated days per actual day on current High Performance Computing Architecture (Stevens et al. 2019; Caldwell et al., n.d.)

(forecast models such as IFS(C. D. Roberts et al. 2018) use approximations to achieve longer timesteps which allow an order of magnitude higher throughput, but these approximations are debated for climate applications(Stevens et al. 2019) ). Allowing for the historical exponential growth of 100 times per decade in computing capacity to continue for the next 20 years (itself a debate (Theis and Wong 2017)), and assuming model performance could scale perfectly to utilize all additional performance, it would require until 2040 for 3 km convection permitting models to achieve the CMIP6-class throughput of 1-100 simulated

years per wallclock day. Global convection resolving models at a 1km resolution might be expected to be 10-100 times slower than convection permitting models (barring unforeseen advances), hence they should not be expected to produce CMIP6-style run lengths and ensemble sizes even by 2040.

As such, a near-term singular focus on *any* single axis of climate model development would come at the expense of significant gaps in knowledge and in our ability to perform climate risk assessments, and in our capacity to have confidence in the range

of possible climate mitigation outcomes. Computational and human limitations dictate that not all of these goals can be fulfilled with the same type and resolution of model.

However, coarse-scale Earth system models continue to have long-standing systematic errors (Stouffer et al. 2017; Eyring et al. 2021) and simulate a large spread in effective climate sensitivity (Schlund et al. 2020), mainly due to unresolved subgrid-scale processes (e.g., clouds and convection). New approaches exist to improve the models with Machine learning (Eyring et

al. 2021; Gentine et al. 2018; Reichstein et al. 2019), (Schlund et al. 2020). Short simulations from global high-resolution climate models might be of limited use in isolation for the projection of future climate evolution and risks, but together with observations, they can serve as information to develop ML-based parameterizations that are then incorporated into coarser-scale Earth system models with demonstrated improved performance compared to the original Earth system model (Gentine et al. 2018). The development of hybrid (physics + ML) Earth system models could thus ideally complement the approach

proposed here, enabling large run-lengths and ensemble sizes needed for robust climate risk assessments that complement km-scale climate modeling activities..



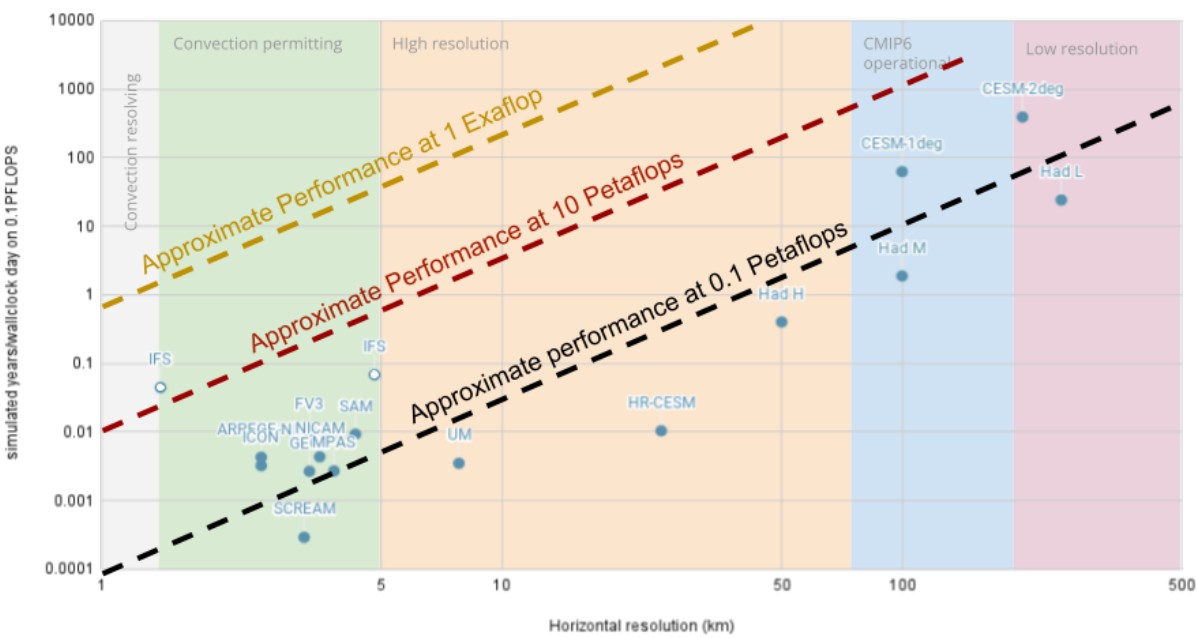

**Figure 6: Illustration of the trade-off between resolution and simulated years using current and projected future computational capacity. Horizontal axis shows the horizontal resolution, estimated as the square root of the area of the largest gridcell, vertical axis shows the simulated years per wallclock-day as reported from operational flagship simulations (see Table B1.). Climate models are filled circles, weather forecast models are open circles. Dotted black line shows the fitted linear relationship between log(resolution) and log(throughput). Red and gold dotted lines show transposed fit for a 100 and 10,000 times increase in computational power respectively (assuming perfect scaling of model performance).**

Here we argue that the current 50-100km resolution 'CMIP' class of climate and Earth system models should continue as a pillar of climate information in parallel to high resolution activities. This class of model would be of greater value if scenarios were, by default, conducted in emissions-driven mode for those models with the technical capacity.

### 4.2 IAMs and scenario development

Emissions-driven simulations to date in CMIP have been highly idealized (e.g. ZECMIP(Chris D. Jones et al. 2019)). An emissions-driven focus allows coupled system processes to be represented in policy relevant scenarios, but this requires a refinement in the way that scenarios have traditionally been framed and categorized (O'Neill et al. 2016). In emissions-driven mode, ESM simulated concentrations, radiative forcing, and temperature will differ from the original IAM simulation (Figure 7c). As such, emissions-driven scenarios should be categorized in terms of their policy strategies (e.g, net-zero dates and land use strategy), not backcasted from pre-defined climate outcomes, a fundamental shift in how scenarios are categorized. In practice, the policy strategies implemented internally in IAMs would still be informed by a climate outcome (e.g., Paris



compliant scenarios), perhaps assessed using a simple climate model - but process uncertainties represented within the downstream ESM ensemble simulation may illustrate that some policies targeted at a given warming level are more robust than others (e.g. scenarios which rely heavily on afforestation which may or may not achieve desired carbon outcomes in all ESMs) or may have different negative impacts on other aspects of the global environment (e.g. air quality or food production capacity).

It is notable that some IAMs already contain process-based land surface models to inform land use emissions estimates (Stevanović et al. 2016). A key distinction in the emissions-driven framework would be that land use transitions (in addition to fossil $CO_2$ emissions), are provided by the IAM system - allowing a *diversity* of land use emissions to be simulated in the ESM ensemble (rather than the status quo where a single set of land use emissions are computed by the IAM) thus modeling the uncertainty in climate implications of land-use transitions.





**Figure 7: Stylized illustrations of the historical (a,b) and proposed (c,d) information flow for CMIP. (a) shows concentration-driven modeling pipeline with prescribed aerosols common in CMIP3 (b) shows concentration-driven modeling pipeline with interactive aerosols common in CMIP5,6 (c) a proposed scenario pipeline for emissions driven simulations in CMIP7 with carbon emissions but maintaining concentration definitions for non-CO2 greenhouse gases (d) a proposed CMIP8 pipeline, with complete emissions driven configuration and process based implementation of CDR and SRM approaches**



## 4.3 A coupled climate-carbon ESM representation for CMIP7

We argue that carbon-climate interactions and feedbacks are central to how the coupled Earth system will evolve in the future
and therefore need to be central to CMIP activities going forwards rather than an optional extra. For CMIP7, this requires that carbon emissions and land activity driven simulations become the default for those models which are capable. ESMs in this configuration require the ability to represent anthropogenic carbon emissions from fossil fuels and land use change and management in the context of a closed and stable carbon cycle, which represents oceanic and land-based sinks. For these models, CMIP7 historical and scenario experiments could be driven by fossil carbon emissions and land use transitions. For
ESMs without the capacity or desire to run in an emissions-driven configuration, scenarios based on simple climate models could still be computed in the conventional ScenarioMIP structure, with guidance that the concentration pathway represented within ScenarioMIP is only one potential outcome of climate policies in terms of emissions, atmospheric concentrations, and climate and carbon cycle responses. Alternatively, non-ESM AOGCMs could be driven by small ensembles of plausible concentration pathways, sampling a range of plausible carbon cycle uncertainty.

Participation in CMIP by models with heterogeneous complexity is not unprecedented. In CMIP5 (Taylor, Stouffer, and Meehl 2012)and CMIP6 (Eyring et al. 2016), only some models were capable of processing aerosol emissions (including aerosol-cloud interactions and feedbacks on natural aerosol emissions such as biomass burning, dust and sea spray) while those without interactive aerosol schemes were driven by predefined loadings. In CMIP3 (Meehl et al. 2007), there was a similar coexistence between models with a thermodynamic slab ocean and those with a fully dynamic ocean (though slab oceans were abandoned
in CMIP5). These periods of coexistence of model complexity proved a necessary and very successful compromise to allow this diversity on the path towards a successful transition to increased complexity across the CMIP ensemble. We argue that now is the right time for the next planned transition to emissions-driven modelling capability.

## 4.4 Towards comprehensive mitigation modeling in CMIP8 and beyond

There are a number of highly informative model developments that are likely too ambitious for the CMIP7 timeline, but are
necessary for a comprehensive process-driven representation of the outcomes of mitigation strategies.

### 4.4.1 Closed cycles for water and other major greenhouse gases

Non-$CO_2$ forcers play a significant role in mitigation dynamics and carbon budget uncertainties, both in terms of forcing and scenario uncertainty (Rogelj et al. 2015). However, the capacity of current generation Earth System Models to produce closed and stable cycles for non-$CO_2$ greenhouse gases lag behind that of carbon dioxide (Séférian et al. 2020), where capacity has
already been demonstrated to replicate historical and scenario simulations in CMIP6 (Arora et al. 2020). While interactive treatment methane (Heimann et al. 2020; Folberth et al. 2022) and nitrous oxide (Xu-Ri et al. 2012) are being developed in Earth System Modeling platforms, no models in CMIP6 yet resolved closed cycles for these gases (Séférian et al. 2020). As



such - pragmatically, on a timescale of CMIP7, there will remain elements of historical and future simulations which will, for most models, remain exogenously defined.

Closing carbon and nitrogen budgets would require a dedicated joint effort in land and ocean model developments and calibration, and inclusion of potentially absent processes such as lateral transport of dissolved organic carbon and nitrogen (Lauerwald et al. 2017; Lacroix et al. 2021) representation of the coastal ocean dynamics (Mathis et al. 2022), and erosion of coastal permafrost (Nielsen et al. 2022). Similarly, models do not currently close the water cycle. Ice sheets and inland glaciers are a dominant component of sea-level rise (itself perhaps the most critical long term climate adaptation challenge

(Hauer et al. 2019)), and yet ESMs do not operationally represent them in coupled simulations. Given this, a number of models have a prioritized focus on including ice sheets and glaciers to "close" the global water cycle (R. S. Smith et al. 2021; Lofverstrom et al. 2020).

### 4.4.2    Assessment of uncertainty in historical and future land use emissions

A more comprehensive, accurate, and consistently-diagnosed representation of historical land-use emissions and processes is

necessary to address both the ensemble bias towards low historical land use emissions as compared to Global Carbon Project estimates in CMIP6 (P. Friedlingstein et al. 2022) and the need for a counterfactual no-land-use scenario found in (Liddicoat et al. 2021). For future scenarios, in order to make ESM simulations consistent with scenario narratives, a greater focus will be required to ensure that IAM internal representations of land use emissions are at least representative of those obtained from the ESMs (though some spread around the central estimate is a desired outcome of the experimental design).

### 4.4.3    Process-based representation of carbon removal and storage

The objective to interactively resolve the processes associated with carbon removal within the structural framework of Earth System Models is a key requirement to providing process uncertainty in carbon dioxide removal (Psarras et al. 2017) . Although isolated ESMs have already been used to investigate the potential effectiveness of removals through Bioenergy

Carbon Capture and Storage (Muri 2018; Melnikova et al. 2022; Kato and Yamagata 2014), or potential oceanic CDR approaches through ocean alkalinization (Fröb et al. 2020) or algal cultivation (J. Wu, Keller, and Oschlies 2023), these capacities remain experimental, and lack representations and accounting of sequestered carbon in emissions-driven simulations. Further, the pipeline for representing the respective amount of CDR carried out in any given IAM scenario and translating this into a consistent set of instructions for an Earth System Model does not yet exist (see Figure 5c/d). As such,

without a dedicated and immediate effort, it is likely that for CMIP7, CDR will also remain exogenously defined in the coming cycle. However, constructing and standardizing these information pipelines for specifying process-based CDR methods in ESMs from IAM output during the CMIP7 timeframe will be needed to create a foundation for fully process-resolved emissions-driven mitigation scenarios beyond CMIP7.



## 5 A proposal for an ESM-DECK

Here we propose a set of experiments for a CMIP7 emissions-driven DECK to be conducted by models in emissions-driven configurations, providing information on relevant climate response diagnostics. The set of experiments is self-contained and not conditional on the existence of a concentration-driven model spinup, but the protocol would be complementary to the concentration-driven experiments in existing CMIP6 DECK (detailed in Table A2). These idealized emissions-driven experiments would allow calculation of key carbon-climate metrics needed to inform climate policy tools such as the IPCC remaining carbon budget for climate stabilization, thus complementing existing concentration-driven metrics.

Table 1 and Figure 8 illustrate a proposal for a set of diagnostic emissions-driven experiments which would provide emissions-driven estimates of TCRE and ZEC, together with objective assessments of climate reversibility under negative emissions. These are illustrated in Figure 7, which uses a diverse ensemble of parameter configurations of the FaIR (C. J. Smith et al. 2018) simple climate model to illustrate the relationship between carbon emissions, concentrations and temperature response under climate uncertainty.

| experiment | mode | CMIP6 MIP | Forcing | branches from | Relevance | Metrics provided from simulation |
|---|---|---|---|---|---|---|
| esm-PiControl | e-driven | DECK | 1850 constant | esm-PiControl-spinup | Stable control climate for e-driven climate | - |
| esm-PiControl-spinup | e-driven | DECK | 1850 constant | - | Pre-equilibrated spinup stage for ESM configurations | - |
| esm-flat10 | e-driven | - | fixed CO2 emission rate (10GtC/yr) for at least 150 years to ensure 2x CO2 concentrations are reached | esm-PiControl | Emissions-driven estimate of TCRE, reaches exactly 1000PgC in 100 years | TCR, TCRE |
| esm-flat10-zec | e-driven | - | zero emissions branching from flat10 in year 100 | flat10 | Idealized calculation of ZEC from flat10 expt, branch in year 100 | ZEC50, ZEC100 |
| esm-flat10-cdr | e-driven | - | Linearly declining emissions by 2GtC/decade from 10GtC/yr (year 100) to -10GtC/Yr (year | flat10 | Idealized calculation of climate reversibility under negative emissions, branching from flat10 | TNZ, TR1000, TR0, tPW |



| | | | 200). Constant -10GtC/yr (years 200-300) | | experiment. | |
|---|---|---|---|---|---|---|
| *esm-CDR-pi-pulse* | e-driven | CDRMIP | 100PgC instantaneous emission (year 0) | esm-PiControl | Provides clean calibration data for climate emulators | |
| *esm-Historical* | e-driven | CMIP | historical emissions | esm-PiControl | Provides historical climate assessment and initial states for e-driven scenarios | - |

**Table 1: A proposal for an "ESM-DECK", which would provide emissions-driven dynamics of the global climate system.**

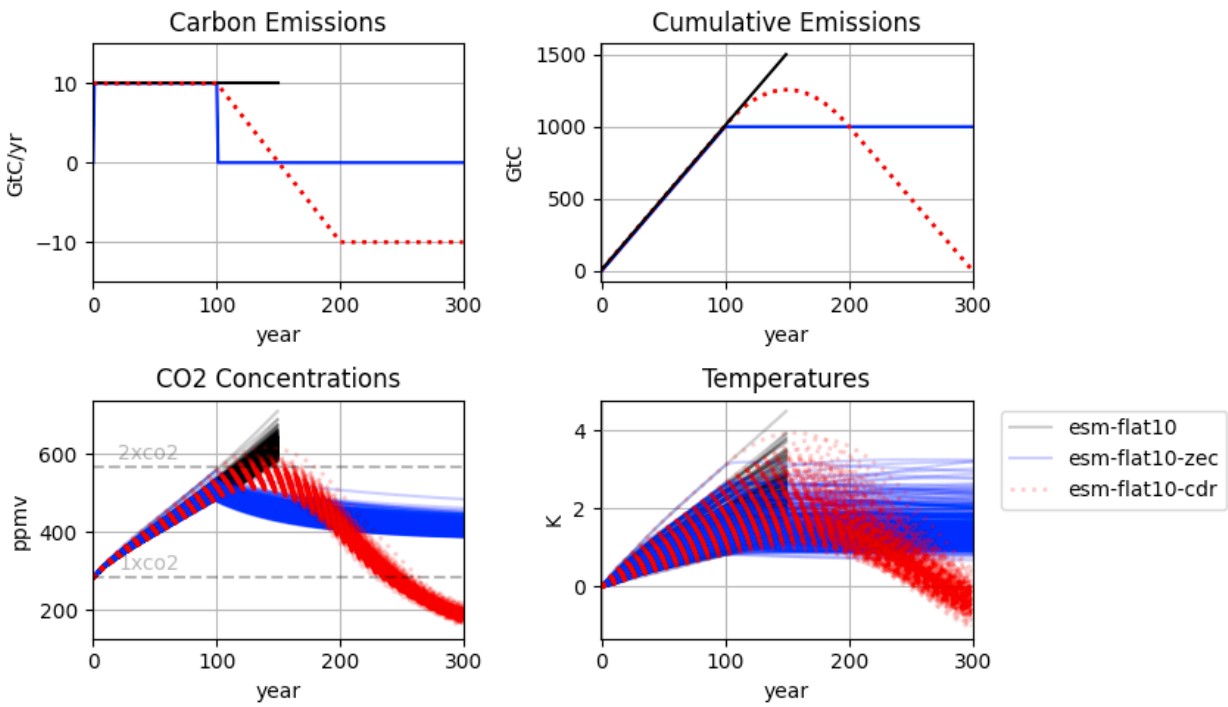

Figure 8: emissions, cumulative emissions, CO₂ concentrations and Global Mean temperatures in the 'flat10' experiments (esm-flat10, esm-flat10-zec and esm-flat10-cdr) as simulated in a perturbed parameter ensemble of the simple climate model FaIR (C. J. Smith et al. 2018).

## 5.1 Transient metrics

The Transient Climate Response (TCR) is defined as the warming in *1pctCO2* at the time of doubling of pre-industrial concentrations(Gregory and Forster 2008), while the TCRE (Matthews et al. 2009) is defined as the ratio of the TCR to the cumulative fossil fuel emissions at the time of doubling (usually expressed as warming per Exagram of cumulative carbon



emissions i.e. per 1000PgC). In CMIP6 (and prior phases), the *1pctCO2* simulation, which increased atmospheric concentrations of CO2 by 1% per year from pre-industrial levels, was used to calculate two metrics of transient climate change. This experiment would consider a constant annual flux of 10PgC of carbon for 100 years (such that the warming after 100

years would correspond to 1000PgC of cumulative emissions - as such, a direct measure of TCRE). Unlike for *1pctCO2*, compatible emissions do not need to be computed and the TCRE can be easily calculated as a time average in the experiment. Figure 9 illustrates that TCRE derived from *1pctCO2* and *esm-flat10* are highly correlated in the simple climate model ensemble, with R=0.983 in the FaIR simple climate model ensemble. We note that the greatest outliers are models with a particularly large zero emissions commitment. If FaIR configurations with a Zero Emissions commitment outside of the IPCC

AR6 assessed range (Lee et al., n.d.) (where the absolute value of ZEC50 was assessed as likely to be less then 0.3K) are excluded, the correlation of TCRE from *esm-flat10* and *1pctCO2* improves further (R=0.987).

We propose that *esm-flat10* could provide an emissions-driven assessment for both TCRE and TCR which would complement *1pctCO2* in CMIP7 while providing a clean experiment which can be branched to assess zero emissions commitment and climate reversibility. In future CMIP phases, as a greater fraction of models adopt default emissions-driven configurations, it

would provide a method for assessment of TCR and TCRE which does not require a concentration-driven model spinup.





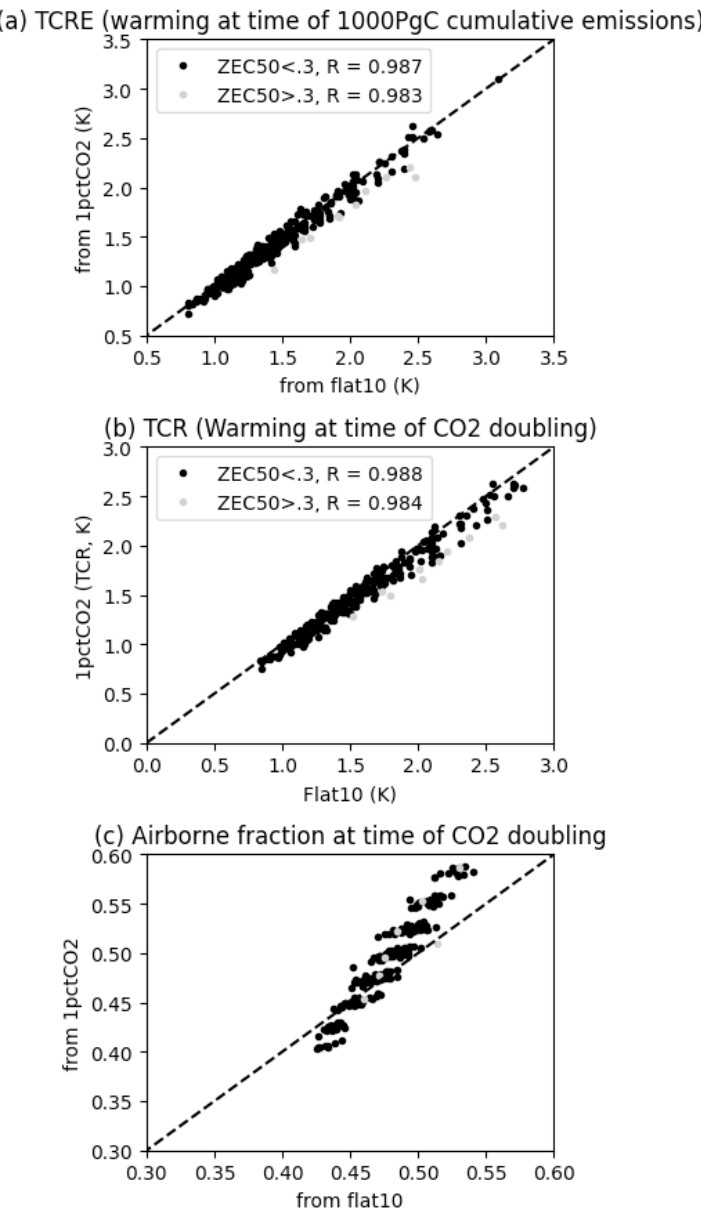

**Figure 9: Comparative calculations of (a) TCRE, (b) TCR and (c) Airborne Fraction from esm-flat10 experiments and the CMIP6 1pctCO2 experiment as simulated in a perturbed ensemble of the simple climate model FaIR** (C. J. Smith et al. 2018)**.**

*Esm-flat10* as a default diagnostic for TCRE would have a number of desirable properties: (1) emissions are constant for all models considered (rather than varying by model under *1pctCO2 - see Figure A1*), (2) emissions are constant throughout the simulation (rather than weighted towards the end of the simulation in *1pctCO2*), (3) peak emission rates are more consistent with those of ambitious climate mitigation scenarios than the diagnosed peak emission rates in *1pctCO2* at the point of reaching 1000PgC cumulative emissions are. *Esm-flat10* also provides a good estimate for TCR by considering warming at the time of



CO$_2$ doubling, where we find a correlation of 0.984 between TCR from *1pctCO2* and *esm-flat10* (similarly improved to 0.994 if only models using configurations with ZEC50 in the IPCC assessed range are considered). CO2 concentration doubling occurs at a later point in time in *esm-flat10,* (between year 110 and year 140 in the FaIR *esm-flat10* ensemble, compared with year 70 in *1pctCO2*), and we find both a generally slightly higher value of TCR in *esm-flat10* than *1pctCO2,* and that the airborne fraction at the time of CO2 doubling is generally higher in *1pctCO2* than *esm-flat10* as a result of this longer time to 640 CO2 doubling.

## 5.2 Zero Emission Commitment metrics

The zero emissions commitment is a measure of the path-dependence of the temperature to cumulative emissions relationship (Koven, Sanderson, and Swann 2023), an estimate of the subsequent global warming that would result after a period of anthropogenic emissions, once they are set to zero (Chris D. Jones et al. 2019; MacDougall et al. 2020). ZECMIP (Chris D. 645 Jones et al. 2019) contains a number of experiments to quantify this behavior, most predominantly with the *esm-1pct-brch-1000PgC* experiment, which branched from the concentration driven *1pctCO2* at the point at which 1000PgC of cumulative emissions had been emitted.





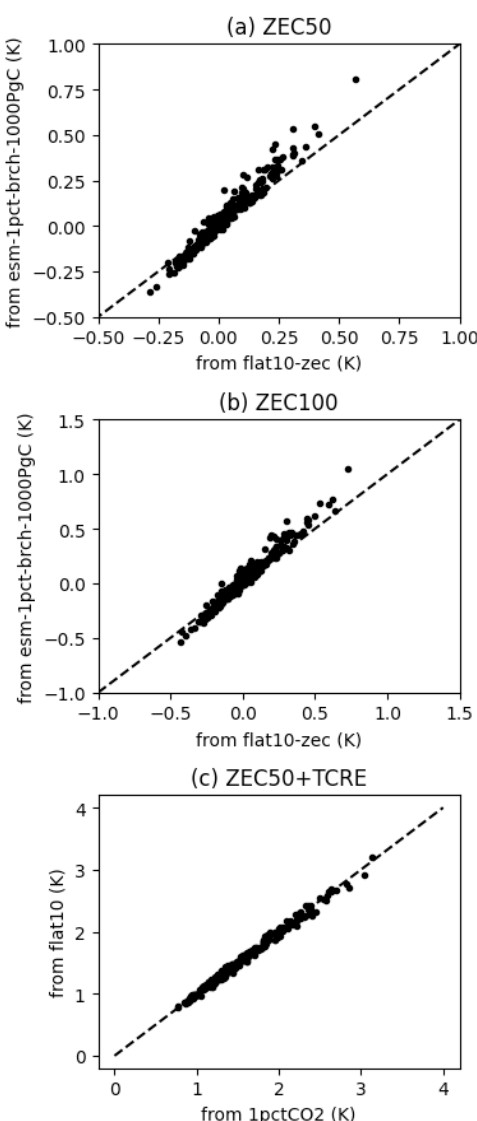

**Figure 10: Computation of Zero Emission Commitment metrics (a) ZEC50, (b) ZEC100 in esm-flat10-zec and esm-1pct-brch-**
**1000PgC as simulated in a perturbed ensemble of the simple climate model FaIR** (C. J. Smith et al. 2018)**, while (c) shows the sum of**
**TCRE and ZEC50.**

Here, we propose a completely emissions-driven alternative derivation for ZEC behavior. *esm-flat10-zec* allows for

computation of temperature changes after an immediate cessation of emissions, similar to the ZEC concept assessed in (Chris

D. Jones et al. 2019). Figure 10 shows that although ZEC50 is highly correlated between the esm-flat10 experiments and the

ZECMIP protocol, we see significant differences in the derived magnitude due to the greater weighting of emissions towards

the end of the experiment in *1pctCO2* - with values of ZEC50 order 50% greater when calculated in *esm-1pct-brch-1000PgC*

(see Figure 10a). Differences in ZEC100 are smaller (around 10%) between the two experiments (Figure 10b).



We can understand these differences due to the partitioning of warming between TCRE and ZEC in the two experimental designs. This is evident in Figure 10c - where we show that TCRE+ZEC50 is near-identical for both methodologies. The tendency for emissions to be weighted towards the latter years of the simulation in *1pctCO2*, as well as the shorter total time period over which emissions occur in *1pctCO2* (~70 vs 100 years), means that models with a higher ZEC50 have a greater fraction of unrealized warming at the time the 1000PgC threshold is exceeded in the *1pctCO2* than in the *esm-flat10* case, where emissions are evenly distributed through a longer experiment - such that more warming is realized in year 100 (corresponding to a cumulative emission of 1000PgC).

Esm-flat10-zec would convey a number of both practical and theoretical advantages over *1pctCO2* as a primary diagnostic of Zero Emissions Commitment. (1) Because the experiment is emissions-driven from the outset, it would not require a change in configuration at the branch point, which poses a technical challenge for some models. (2) The branch point is identical for all models (unlike in *esm-1pct-brch-1000PgC,* where the year in which 1000PgC of compatible cumulative emissions is exceeded must be calculated retrospectively to find the appropriate branch year). (3) This common experimental setup would allow the easier automation of ensembles in the calculation of both TCRE and ZEC, without needing to calculate compatible emissions to find the appropriate branch point. (4) The maximum rate of CO2 emissions in *esm-flat10* (10 Pg C/yr, vs ~20 Pg C/yr for *1pctCO2*) is closer to realistic values that are projected for ambitious policy scenarios, where emissions must peak and decline from their present values of ~10 Pg C/yr within decades to achieve Paris Agreement-compatible warming targets. Because TCRE and ZEC should be quantified relative to a consistent set of experiments (Koven, Sanderson, and Swann 2023), and because TCRE as assessed by IPCC AR6 was further decomposed into separately-assessed Transient Climate Response (TCR) and Airborne Fraction (AF) components (AR6 WG1 ch. 5 & 7), using *esm-flat10* and *esm-flat10-zec* as the primary CMIP7 quantification of TCRE and ZEC would thus require a self-consistent quantification of TCR, AF, and ZEC, and in particular a mapping of TCR between the *esm-flat10* scenario and TCR as assessed from multiple other lines of evidence.

### 5.3 Reversibility metrics

An increasing feature of the discussion of future Paris-Compatible pathways is an assessment of the reversibility of the climate system, both in a global sense (Zickfeld et al. 2013; P. Wu et al. 2015) and in terms of regional and subsystem responses (Armour et al. 2011; Martin et al. 2022). In CMIP6, a number of idealized experiments were conducted under CDRMIP (Keller et al. 2018) which included a concentration-driven extension of *1pctCO2* called *1pctCO2-cdr*, which prescribed a 1% rampdown in concentrations at the point at which *1pctCO2* reached quadruple pre-industrial levels. This experiment undergoes a large discontinuity in compatible emissions at the transition from upwards to downwards branches, making it less useful as an indicator of realistic transitions to negative emissions (see Figure A1) (Koven, Sanderson, and Swann 2023).

Here we propose an emissions-driven extension to *esm-flat10* to address this need: *esm-flat10-cdr* would serve as an emissions-driven idealized experiment to assess the dynamics of climate reversibility under reducing emissions and net-negative emissions. The experiment would allow for a number of simple idealized diagnostics which would be relevant to the net zero transition and the response of the system to net negative emissions. *Esm-flat10-cdr* would branch from *esm-flat10* in year 100,



after 1000PgC of emissions, ramping down emissions linearly over 100 years from +10PgC/yr to -10PgC/yr and then maintaining a negative flux of -10PgC/yr for an additional 100 years.

This *esm-flat10-cdr* experiment would provide a number of advantages over *1pctCO2-cdr*: (1) an emissions-driven metric of climate reversibility with a continuous emissions timeseries, (2) an idealized net-zero transition to measure the lags in the

climate system in the decades around net-zero as emissions pass from net positive to net negative, (3) characterization of asymmetries in the climate response relative to emissions rather than to concentrations, by using a symmetric and continuous reversal from positive to negative $CO_2$ emissions, and (4) initial emissions and a decarbonisation rate which are comparable to an aggressive mitigation scenario. These features are all also present in the gaussian cumulative emissions experiment described by (Koven, Sanderson, and Swann 2023) and listed as *esm-restoration* in table A1 and figure A1, which also features

an asymptotic rise in emissions at the start of the industrial period and an asymptotic tapering of negative emissions to zero as cumulative net zero emissions is achieved. The key advantage of *esm-flat10-cdr* over *esm-restoration* for an ESM-DECK is that it aligns well with *esm-flat10* and *esm-flat10-zec* to form a coherent set of interrelated experiments and metrics.

We propose that *esm-flat10-cdr* could be used to define a number of reversibility metrics:

1. **tPW** - the time difference between the peak value of 20-year smoothed global mean temperatures and the point that
net zero is achieved in *esm-flat10-cdr* (year 150). This metric has a clear policy-relevant translation as the expected time it will take for the climate system to achieve maximum $CO_2$-driven global warming after (or before) reaching net zero emissions under a smooth positive-to-negative emissions transition.

2. **TNZ** - a 20 year average around year 150 in *esm-flat10-cdr* minus a 20 year average around year 125 in *esm-flat10*. TNZ would be the temperature simulated at net zero minus the expected temperature at net zero using cumulative
emissions proportionality. It represents the degree to which temperatures at net-zero might deviate from what would be expected from combining a remaining carbon budget and an estimate of TCRE (Rogelj, Forster, et al. 2019). This could be easily calculated using a combination of the *esm-flat10* and *esm-flat10-cdr* experiments for a cumulative carbon emissions total of 1250GtC. *Esm-flat10-cdr* reaches net zero emissions in year 150, with a cumulative emissions of 1250GtC (calculated from year 0, see Figure 7). *Esm-flat10* itself reaches 1250GtC in year
715 125.

3. **TR1000** - be calculated as a 20 year average around year 200 in *esm-flat10-cdr* minus a 20 year average around year 100 in *esm-flat10*. TR1000 would be a measure of hysteresis in global mean temperature when cumulative emissions return to 1000PgC on the downward branch minus the expectation from TCRE. This could be calculated using a combination of the *esm-flat10* and *esm-flat10-cdr* experiments for a cumulative carbon emissions total of
1000GtC. *Esm-flat10-cdr* reaches 1000PgC cumulative emissions in year 200 on the downward branch (see Figure 7). *Esm-flat10* itself reaches 1000GtC in year 100.

4. **TR0** - a 20 year average around year 300 in *esm-flat10-cdr* minus mean global temperatures in *esm-pictrl*. TR0 would be a measure of hysteresis in global mean temperature when cumulative emissions return to zero after a period of negative emissions. This could be calculated using a combination of the *esm-pictrl* and *esm-flat10-cdr*
experiments. *Esm-flat10-cdr* reaches zero cumulative emissions in year 300 on the downward branch (see Figure 7).

In the FaIR SCM ensemble of *esm-flat10–cdr*, we see some strong emergent relationships between these reversibility metrics and ZEC metrics (see Figure A2)—consistent with similar reported metrics derived from CMIP6 ESMs (Koven et al. 2022) and FaIR (Koven, Sanderson, and Swann 2023)—but noting the structural assumptions in FaIR do not allow for threshold





behavior which may be present in ESMs. As such, *esm-flat10-cdr* would provide an assessment - beyond ZEC, of underlying nonlinearities and irreversibilities in Earth System Models.

## 5.4 Pulse response and equilibrium climate sensitivity

Finally, we propose the inclusion of an idealized experiment following (Joos et al. 2013), to inform the Impulse Response Function given a 100GtC emissions pulse of CO2 into the atmosphere (See Figure A1). Although such an experiment is clearly
idealized, the experiment has uses in the clean calibration of simple climate models (Schwarber et al. 2019), which are regularly used in assessment as climate emulators. A similar experiment was included in CMIP6 under the CDRMIP protocol *esm-CDR-pi-pulse.*

Notably, an ESM Deck in isolation would not allow for the direct calculation of equilibrium climate sensitivity (ECS) which is the equilibrium temperature response to a doubling of atmospheric carbon dioxide concentrations, though it would be
desirable to know ECS to enable comparison of climate feedbacks across the wider CMIP7 ensemble. The ECS could be inferred indirectly by calibrating a simple climate model to the ESM output of *esm-CDR-pi-pulse,* and assessing the climate feedback parameters. Alternatively, for centers with a spun-up emissions driven configuration, a short concentration-driven pre-industrial control could be produced relatively cheaply by branching from *esm-picontrol* holding $CO_2$ concentrations constant at the model's equilibrated average level (rather than prescribed pre-industrial levels), preventing the need for a long
additional spinup. The CMIP6-style DECK experiments *abrupt4x* and *1pctCO2* could then be conducted with concentrations relative to the model's own equilibrium pre-industrial $CO_2$ concentrations to determine ECS and TCR.

## 6    Conclusions

Future climate scenarios have been primarily framed in terms of concentrations (or in terms of metrics of global warming) since the Special Report on Emissions Scenarios (SRES) was introduced (Nakicenovic et al. 2000) at the turn of the 21st
century. More recently, a 'parallel process' (Moss et al. 2010) advocated defined concentration pathways, with climate effects conditional on concentration pathways assessed by Earth System Models while Integrated Assessment Models explore scenarios consistent with the pathways. This approach was chosen pragmatically to allow the two communities to work concurrently, and because only a subset of Earth System Models have operationally incorporated interactive and closed carbon cycles. However, this framing does not allow carbon cycle uncertainty as represented by diverse, process-resolving Earth
System Models to be manifested in the scenario outcomes, thus omitting a dominant source of uncertainty in meeting the Paris Agreement (Chris D. Jones and Friedlingstein 2020; Holden et al. 2018).

In addition, a rapidly evolving policy landscape increasingly requires information to differentiate between scenarios which represent both different levels of mitigation ambition and different mitigation strategies. A decade earlier in the timing of net-zero $CO_2$ represents a huge economic investment (Nieto 2022), but at present we do not have scenario outcomes to clearly
illustrate the associated climatic benefits in a way that accounts for all uncertainties. Thus, there is no direct and self-consistent



simulation of the benefits of mitigation which can be associated with incremental reductions in emissions. On the implementation side, national mitigation policies that (explicitly or implicitly) rely on land use and carbon dioxide removal (CDR) techniques introduce significant uncertainties which remain unsampled in the current ESM scenario framework.

The utility of ESMs is to a large degree shaped by how they are deployed in model intercomparison projects. For example, it

has been argued that ESMs can be made more relevant to climate adaptation challenges by resolving and outputting relevant human and ecosystem climate impacts (Bonan and Doney 2018). Similarly, with the right experimental design, many existing ESMs already include components that can provide valuable insights into the uncertainty surrounding the timing and implementation of net-zero policies.

The upcoming Coupled Model Intercomparison Project Phase 7 (CMIP7) provides an opportunity to move towards a

framework enabling an operational assessment of emissions-based policies. This would happen through the explicit representation of carbon dioxide emissions in the context of multiple plausible representations of natural climate system feedbacks. This framework will serve as a structure for incorporating the uncertainties associated with the effectiveness of land use and CDR techniques as part of a mitigation portfolio, some of which are already implemented in current-generation Earth System Models, and some of which require further development beyond the timescale of CMIP7. This framework needs

to be flexible enough to accommodate different models at various stages of development, and different configurations focusing on different elements of the climate problem, necessitating a hybrid approach for CMIP7.

We propose that the existing CMIP6 model for accommodating a range of aerosol complexity is extended to the simulation of an emissions and activity-driven carbon cycle. Concentration pathways should still be available for models that require them (and for configurations where carbon cycle feedbacks are not the primary focus, such as high-resolution experiments and some

perturbed parameter ensembles). This will need careful communication in the ScenarioMIP framework, as only a subset of models will be subject to carbon cycle uncertainties (though this remains analogous to the CMIP6 treatment of aerosols, where only some models process aerosol emissions directly). It is expected that some climate-relevant forcers such as nitrous oxides and methane will not be represented interactively by a large fraction of models on the timescale of CMIP7, thus exogenous concentrations will still be required in most cases.

Looking ahead to CMIP8 and beyond, ESMs will continue to occupy a critical niche, maximizing the representation of human actions involved in climate mitigation and adaptation in a risk framework which relies on deep and diverse process understanding which is uniquely represented in the collective historical and ongoing effort encapsulated in the CMIP ensemble. Future efforts (and their associated computational expense) should be focused on areas where they can add the most value to understanding the Earth system in an ever-widening ecosystem of simple and complex model configurations which are

increasingly well adapted to different aspects of the climate problem.

We argue that a better understanding and representation of emissions-driven dynamics remains one pillar of a wider effort needed to adapt Earth System Models to evolving climate challenges. It has been documented already that there is a need for physically realistic, higher resolution model output (Schär et al. 2020; Bauer, Stevens, and Hazeleger 2021), but these must be supplemented by lower resolution operational configurations which are capable of simulating large initial conditional and



parametric ensembles of century driven global response to diverse mitigation strategies. Machine learning may also change this tradeoff - approaches are currently being explored to improve the representation of key resolution-dependent physical processes in global climate models (Gentine et al. 2018), with encouraging results. Such approaches also hold great potential for better utilizing observations to inform future improvement of carbon cycle processes in ESMs (Forkel et al. 2019). Bringing together ML developments for both the physical and carbon-cycle components of future emission-driven ESMs offers the

potential for a major advance in our ability to model the coupled global climate and carbon cycle (Eyring et al. 2021). However, there remain conceptual problems with overreliance on machine learning for century scale projections where no training data is available (D. Watson-Parris 2021)

By shifting to primarily emissions-driven simulations, the ESM ensemble would become a critically relevant part of the scenario assessment framework, providing the best available process-based estimations of the distribution of potential

outcomes resulting from proposed societal transformation pathways. A scenario which achieved a set of policy goals based on the prior generation of models may not achieve those same outcomes with updated models. A default emissions-driven scenario infrastructure would make such comparisons transparent, making it clear when developments in process understanding have measurable impacts on the projected risk associated with a given mitigation strategy.





## Appendix A1 – Additional idealised experiments not in the main ESM-DECK proposal



**Figure A1: Simple model ensemble illustrations of various idealized experiments relevant to the quantification of emissions-driven Earth System dynamics, using perturbed variants of the FaIR** (C. J. Smith et al. 2018) **with parameters sampled as in** (Koven, Sanderson, and Swann 2023). **Columns from left to right indicate carbon emissions as a function of time (starting from an equilibrated pre-industrial state), atmospheric CO2 concentrations and surface temperatures. The top two rows show concentration driven idealized experiments (with the exception of esm-1pct-brch-1000PgC, which switches from concentration driven to emissions driven when 1000PgC cumulative emissions are exceeded) where compatible emissions are calculated for each ensemble member, while the bottom three two rows show emissions-driven experiments.**

| experiment | mode | CMIP6 MIP | Forcing | branches from | Relevance |
|---|---|---|---|---|---|
| esm-1pct-brch-1000PgC | e-driven | ZECMIP | zero emissions | 1pctCO2 | Follows CMIP6 ZECMIP protocol, branches from 1pctCO2 (for most models), |




| experiment | mode | CMIP6 MIP | Forcing | branches from | Relevance |
|---|---|---|---|---|---|
| | | | | | produces non-continuous emissions |
| esm-bell1000PgC | e-driven | ZECMIP | gaussian emissions profile, totaling 1000PgC at end of experiment | esm-PiControl | Follows CMIP6 ZECMIP protocol, branches from piControl (for most models), continuous emissions |
| esm-pulse-reversal | e-driven | - | 100PgC instantaneous removal (year 100) | CDR-pi-pulse | Provides information on reversibility of pulse injection of carbon |
| esm-restoration | e-driven | - | Gaussian derivative curve(Koven, Sanderson, and Swann 2023). Cumulative emissions are zero at end of simulation | esm-PiControl | Provides climate system lags and policy relevant departures from TCRE-like behaviour under an idealized net-zero transition |

**Table A1: Additional idealized emissions-driven simulations not included in the main recommendation in Table 1.**


| experiment | mode | CMIP6 MIP | Forcing | branches from | Relevance |
|---|---|---|---|---|---|
| PiControl | c-driven | DECK | 1850 constant | PiControl-spinup | Provides (ideally stable) control climate |
| 1pctCO2 | c-driven | DECK | 1pct annual CO2 concentration ramp from 1850 | PiControl | Provides TCR and TCRE, but uses compatible emissions which make TCRE estimate less relevant to real world response to constant emissions |
| abrupt4xCO2 | c-driven | DECK | Instantaneous concentration quadrupling from 1850 | PiControl | Compute ECS - questions due to assumed radiative balance and sensitivity of gregory result to end of simulation |
| Historical | c-driven | CMIP | historical concentrations | PiControl | Provides historical climate assessment and initial states for c-driven scenarios |
| CDR-reversibility | c-driven | CDRMIP | 1pct annual CO2 concentration decline from 4xCO2 (end of 1pctCO2 simulation) | 1pctCO2 | Provides thermal reversibility information in c.driven mode - branches from 1pctCO2, highly discontinous compatible emissions timeseries creates transient artifacts |
| abrupt4xtoPI | c-driven | - | 1850 concentrations | abrupt4xCO2, year 140 | Provides information on thermal asymmetry in warming and cooling response to step changes in forcing |

**Table A2: Additional idealized concentration-driven simulations not included in the main recommendation in Table 1.**



## Appendix B – reference information on models and reported HPC performance

| Model | Resolution | Cores used | GFLOPS/core | PFLOPS used | Simulated days/wallclock day (reported) | Simulated days/wallclock days (scaled to 0.1PFLOPS) |
|---|---|---|---|---|---|---|
| ARPEGE-NH | 2.5 | 7200 | 23.1 | 0.17 | 2.6 | 1.56E+00 |
| FV3 | 3.3 | 13,824 | 38.4 | 0.53 | 19 | 3.58E+00 |
| GEOS | 3.3 | 20,480 | 31 | 0.63 | 6.2 | 9.77E-01 |
| ICON | 2.5 | 12,960 | 40 | 0.52 | 6.1 | 1.18E+00 |
| IFS | 4.8 | 12,960 | 38.4 | 0.5 | 124 | 2.49E+01 |
| MPAS | 3.8 | 9216 | 38.4 | 0.35 | 3.5 | 9.89E-01 |
| NICAM | 3.5 | 2560 | 64 | 0.16 | 2.6 | 1.59E+00 |
| SAM | 4.3 | 4608 | 38.4 | 0.18 | 6 | 3.39E+00 |
| UM | 7.8 | 12,240 | 38.4 | 0.47 | 6 | 1.28E+00 |
| SCREAM | 3.2 | 217.6 | 45 | 4.7 | 5 | 1.06E-01 |
| Had L | 250 | 9000 | 38.4 | 0.02 | 1460 | 8.80E+03 |
| Had M | 100 | 3600 | 38.4 | 0.07 | 474.5 | 6.86E+02 |
| Had H | 50 | 1800 | 38.4 | 0.12 | 182.5 | 1.47E+02 |
| HR-CESM | 25 | 23404 | 38.4 | 0.9 | 730 | 3.80E+00 |
| CESM-1deg | 100 | 1800 | 38.4 | 0.07 | 5110 | 2.28E+04 |
| CESM-2deg | 200 | 900 | 38.4 | 0.03 | 8760 | 1.43E+05 |
| IFS | 1.4 | 138240 | 166 | 22.95 | 62.05 | 1.62E+01 |

**Table B1: Data for Figure 2, showing reported model throughput and number of CPU cores used for low and high resolution configurations** (Stevens et al. 2019; Dueben et al. 2020; Caldwell et al., n.d.; Small et al. 2014; M. J. Roberts et al. 2019; Chang et al. 2020; "CESM1 Timing Table" n.d.)**. Core performance is taken from advertised peak CPU single float performance on the corresponding machine.**



**Code availability**

All code to reproduce Figures in this study is archived at https://zenodo.org/record/8349377.  The FaIR simple climate model used to simulate ESM-DECK experiments is available at https://github.com/OMS-NetZero/FAIR

**Data availability**

CMIP6 model output is available through the Earth System Grid Foundation (ESGF).  CMIP6 scenario data is available at https://greenhousegases.science.unimelb.edu.au/ and https://tntcat.iiasa.ac.at/SspDb/ .Global Carbon Budget data is available at https://www.icos-cp.eu/science-and-impact/global-carbon-budget/2022

**Author contribution**

BMS wrote the first draft of the paper and produced figures.  Additional analysis was carried out by CK. All authors provided
input, comments and editing on the various parts of the analysis. In addition, modeling center representatives (JD,VE,CDJ,CJ,DML,RAF,JL,IRS,TI,RS,SZ) were co-responsible for performing subsets of the CMIP6 simulations considered and publishing their model output to the ESGF.

**Competing Interests**

At least one of the (co-)authors is a member of the editorial board of Geoscientific Model Development.

**Acknowledgements**

BS, VE, RS and RF acknowledge funding by the European Union's Horizon 2020 (H2020) research and innovation program under Grant Agreement No. 101003536 (ESM2025 – Earth System Models for the Future) and  821003 (4C, Climate-Carbon Interactions in the Coming Century).  BS and C-FS acknowledge funding from 101003687 (PROVIDE) . VE additionally
acknowledges funding by the European Research Council (ERC) Synergy Grant "Understanding and Modelling the Earth System with Machine Learning (USMILE)" under the Horizon 2020 Research and Innovation program (Grant Agreement No. 855187). CDK acknowledges support by the Director, Office of Science, Office of Biological and Environmental Research of the US Department of Energy under contract DE-AC02-05CH11231 through the Regional and Global Model Analysis Program (RUBISCO SFA) . MG acknowledges funding under European Union's Horizon Europe research and innovation
programme under grant agreement no. 101056939 (RESCUE). NM acknowledges funding from the Emmy Noether scheme by the German Research Foundation 'FOOTPRINTS - From carbOn remOval To achieving the PaRIs agreemeNt's goal:



Temperature Stabilisation' (ME 5746/1-1). ALSS acknowledges support from DOE BER RGMA award DE-SC0021209 to the University of Washington.

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
