# Peer review of "The need for carbon emissions-driven climate projections in CMIP7"

_EGUsphere, 2023_

## Author Response (AR1)

Dear editor,

We have extensively revised this paper for GMD in light of the reviewers' comments - including a restructuring of the text, adding (and removing) several sections and correcting references throughout.

Summary of major changes:

- new section 4, collecting text on limitations and challenges
- new section 4.2 on challenges for detection/attribution
- new section 4.3 on use of emissions driven scenarios in assessment
- revised crop harvest figure with multiple IAMs
- new summary of key benefits of e-driven in the abstract
- consistent use of SSPX-Y notation
- cutting the flat10 ensemble analysis (to be put in a seperate flat10MIP paper)
- cutting the section on high resolution computational burden
- added section on diagnosis of land use emissions and relationships to LUMIP
- revised Figure 5 and added box/whisker plots for temperature changes in e-driven and c-driven CMIP6 scenarios
- updated the 'esm deck' discussion to be more current, in lines with plans for the CMIP7 fast track
- added a 'flat10-nz' scenario to the fast track recommendation, which branches from flat10cdr as it crosses net zero and holds at zero emissions.

Point by point response follows:

**Response to reviewer 1**

*General:*

*I congratulate the author group for pulling together this important and timely paper in the preparation of CMIP7. Please consider my below comments and questions in the hope that they strengthen and clarify the manuscript. Without doubt, this is an important paper and should be published after revisions. Some of the outstanding highlights in my view are for example the list of DECK runs and the definition of reversibility metrics (Lines 700ff).*

Many thanks for the positive assessment!

*Comments:*

*In no particular order, I will here provide a few over-arching comments and questions before adding comments with regard to specific lines of the manuscript.*

*RC1: Limitations & New Challenges:  I read the paper with great interest and it lists many benefits of making the carbon emission-driven experiments a priority in CMIP7. I agree. I was wondering whether the authors could provide more discussion on the new challenges that arise under such a framework, though. I think a new section that gathers some of these challenges or limitations and how they could possibly be addressed would strengthen the manuscript.*

Thanks for this suggestion - we add a new section 4 on limitations in the emissions-driven framework.

*RC1.1: How to deal with additional bias in historical runs? The appropriate reflection of carbon cycle uncertainties in future climate simulations seems to come at the cost of inserting additional bias into historical runs. For example, Figure 5 shows a substantial range of CO2 concentrations by 2020 for the carbon emission driven CMIP6 models. We know what CO2 concentrations were in the past, so any CMIP6 derived historical climate will now be subject to an additional bias – which is not present in the historical concentration-driven runs. The authors seem to briefly make the argument that those differences of +-40ppm do not matter in the bigger scheme of things. Yet, 40ppm approximately equate to 20 years in the historical evolution of CO2 concentrations. Thus, do the authors suggest that for historical D&A studies, one should still use concentration driven runs to avoid that extra error? Or should emission-driven climate output be somewhat bias corrected (e.g. a model that has 40ppm too low CO2 concentrations gets somehow adjusted in terms of its tas, pr etc. output to match actual CO2 concentrations) (and once one is at it, would the authors also suggest an aerosol radiative forcing mismatch bias correction to the extent possible?)? Or would a hybrid approach be worth discussing where ESMs should be CO2-concentration driven for the historical period but then switch over to being emission-driven for the future scenarios? Anyway, I at least suggest some more reflection and detail on that point (see also below comment on Figure 5, line 388).*

Thanks - added a section 4.2 on implications for detection studies:

**"4.2 Attribution in emissions-driven simulations**

Attribution studies in DAMIP (Gillett et al. 2016) and in general rely on a linking cause and effect; where the cause has historically been interpreted as the change in a climate forcer (concentrations of

greenhouse gases, solar or volcanic activity etc), and the effect is some climate impact variable of interest (large scale or regional responses in climate impact variables(Hegerl and Zwiers 2011), or the probability of some specific event (Naveau, Hannart, and Ribes 2020)). Emissions-driven simulations pose an issue for this framework because the concentrations and forcing cannot be directly controlled as inputs to the climate model.

This presents two broad options for attribution studies in an emissions-driven model ecosystem. The first is by changing the framing of the question – to address instead the linkages between emissions and outcomes. For example, whereas the current DAMIP design has a "hist-CO2" experiment, which prescribed historical CO2 concentrations while holding all other forcers at historical levels; an emissions-driven equivalent would instead prescribe historical CO2 emissions. Such a framework would allow a more direct assessment of the outcomes of anthropogenic activity.

A second option is to implement a hybrid approach, where some simulations are conducted either in concentration driven mode, or bias corrected such that emissions are adjusted to recreate historical concentration pathways."

*RC1.2: How will future "multiple-lines of evidence" temperature projections be derived? In IPCC AR6 Chapter 4, for the first time in IPCC history, some weighted CMIP6 ensembles (Lang et al., Ribes et al., Tokarska et al., see Figure 4.11 in IPCC AR6 WG1) were used as one part of the 'assessed' surface temperature projections. The constraints were aiming to constrain and weigh the CMIP6 ensemble in terms of the models' temperature response to deal with the 'hot model' problem in a way (simplified speaking). I am not arguing that it is wrong to fully reflect the carbon cycle uncertainties also in these future projections (quite the contrary), but it does pose the challenge of how to undertake this weighting if now one significant extra chain member is included in the ESM results. For example, suppose a model fails to appropriately produce historical CO2 concentrations, but would nevertheless produce ok temperature hindcasts (thanks to possibly offsetting errors), would that model's temperature projections for the future than be fully included in that CMIP7 ensemble average (even though one knows that probably its future CO2 concentrations are way off?). It would be great to know how the evolution of this Chapter 4 IPCC AR6 WG1 approach to combine multiple lines of evidence for GSAT projections should be adapted or evolve under the carbon-emission driven results. For example, if CMIP7 models would show the same performance to reproduce historical trends, then implied emissions of zero CO2 by 1950 (see e.g. your Figure 2 - or vice versa too high rates of increase in CO2 concentrations, Figure 5) would raise possibly concerns whether future projections are to be trusted?*

Thanks - added the following paragraph:
"

**4.3      Informing multiple lines of evidence**

Longer causal chains from emissions-driven simulations may accelerate a shift away from the use of ESM ensembles in assessment from being an ensemble of opportunity used as a proxy for climate uncertainty. We would argue that this transition has already occurred. IPCC reports up to AR5 relied heavily on ESM ensemble distributions as proxies for climate uncertainty.

However, IPCC AR6 utilised some specific methodologies (Ribes, Qasmi, and Gillett 2021; Brunner et al. 2020) to reweight ESM distributions of simulations conditional on their historical simulated climate change in the context of observations. These methodologies considered primarily the physical response of the climate system to historical concentrations, and were used to address the assessed 'hot model' bias (Hausfather et al. 2022) in which the CMIP6 distribution contained some models which notably simulated historical warming beyond that seen in observations. A shift to emissions-driven simulations would introduce an additional source of potential bias in historical concentrations, which would need proper treatment during any assessment. Any weighting scheme would need to properly represent both biases in physical and carbon cycle elements, together with interactions between those elements (multi-variate approaches exist for treating correlated errors(Sanderson, Wehner, and Knutti 2017) (such as errors in CO2 concentrations and global mean temperature).

"

*RC1.3: Comprehensive reflection of uncertainties across the cause-effect chain. I raise this issue not because it is a problem with the manuscript, but because shifting to CO2-emission and landuse-activity-based scenarios does affect this issue. The key question is: How could IPCC (or the scientific community) get to a comprehensive reflection of uncertainties across the full cause effect chain from human activities and emissions to the biogeophysical results that ESMs provide? (i.e. consider it the general case of RC1.2). The climate services sector is (legitimately) demanding such products, for example, a joint probability distribution of a heat wave at location x with a heavy precipitation event at location y. Let's suppose we have around 50 ESMs, which each are integrating a fully fledged carbon cycle. Suppose that observational constraints on past warming on the one side and past carbon cycle behaviour on the other would render 30% of ESMs in each verification step as 'outside of observations and judged to be unrepresentative of our best expert knowledge'. Then we are left with roughly half of the ESM models (50\*0.7\*0.7=24.5) (assuming independence of how good a model does on past warming and past carbon cycle). However, if there is a way to combine all the 'non-dismissed' carbon cycle behaviours with all the non-dismissed warming*

*behaviours, we could potentially create a more comprehensive reflection of future climate uncertainties. Anyway, shifting to longer analysis chains (i.e. starting with emissions rather than concentrations) within our gold-star tools (ESMs) creates more opportunities for some components to be 'non-ideal' and hence invalidate (invalidate in terms of policy-relevance) the output from components of the ESMs (say: the AOGCM) that would otherwise be 'good' ('good' in terms of 'being a valid/useful/plausible reflection of the earth system on the basis of our current understanding). In other words, the welcomed evolutionary step of 'self-consistency' across the cause-effect chain that the integration of the carbon cycle in the main experiments brings with it, comes at the potential price of 'thinning' the ultimate evidence base (or for having to post-process the ESM results by compartmentalising and re-combining the results of the 'valid' components). I am not suggesting this paper has to solve that issue, but as emission-driven runs with ESMs will accentuate the problem of how an overall reflection of uncertainty across all our best ESM components can be gained, it would be great if the authors can briefly point to their thoughts on the issue. This could be part of a new subsection 'limitations' or 'challenges' (with the other points RC1.1 and RC1.2 above). (PS: and of course, this problem of 'longer chains get more vulnerable' is not at all unique to emission- versus concentration-driven runs).*

This is an interesting point - and quite related to 1.2.  We address them together in section 4.3:

"As the length of the process chain increases, it will become increasingly unlikely that ESMs will simultaneously reproduce the joint historical evolution of emissions, concentrations, and climate response. As such, it might become more useful in assessment to consider ESM ensembles as  being sparse samples in a high dimensional complexity space which is illustrative of potential coupled interactions of the Earth System.  Such an interpretation pairs well with the use of meta-models (Nicholls et al. 2022) which can be used to interpolate in a higher dimensional response space and filter between global scale projections using observations(C. Smith et al. 2024).

"

*RC2: Crop harvest figure 3. The conclusions drawn from this figure do not seem to be fully convincing at first without further explanation. The overall harvest flux uncertainty across ESMs seems enormous. Thus, the absolute levels of the REMIND-MAgPIE simulations do not seem to be the issue as such. However, the discussion seems to suggest that the difference of crop harvest between the SSP5-85 and SSP-34over seems 'at worst completely unphysical' (Lines 201ff). A few questions arise that would be great to clarify for a more compelling case: How do other IAMs perform in this comparison. IMAGE is one model with high BECCS in the past? And how do newer generations of the IAMs fare, when e.g. considering NGFS Phase-4*

*scenarios? Given that the ESMs should theoretically have been forced with the landuse activity patterns that are consistent with the REMIND-MAgPIE results, how come that those patterns do not show up as harvest difference? What exactly is part of the crop harvest flux estimates of the ESMs and how does it differ to the normal crop harvest plus Bioenergy Harvest or Net BECCS flux from REMIND- MAgPIE? Wouldn't it be better to show the aggregate REMIND-MAgPIE crop harvest and bioenergy harvest combined? It would be good to see some REMIND- MAgPIE experts involved in the characterisation of this difference (Jessica Strefler, Alexander Popp, Elmar Kriegler, or others).*

Thanks for this - and good points all round.  We've rewritten this paragraph and simplified/remade the figure with more IAM simulations.  Note we are not including any non-SSP scenarios on the figure, because that detracts from the point of the argument (that comparable scenarios in ESMs and IAMs can have very different harvest productivities).

"We can illustrate in Figure 3 the scale of these potential uncertainties in the feasibility of land-based CDR capacity using a pair of scenarios from CMIP6; the highest emission member of the ScenarioMIP ensemble SSP5-8.5  and the extreme overshoot scenario SSP5-3.4-overshoot (Kriegler et al. 2017; Riahi et al. 2017), which assumes a significant amount of BECCS is deployed in the latter half of the 21st century (with bioenergy crop production of 9PgC/yr  by 2100).  In CMIP6 ScenarioMIP, both SSP5-8.5 and SSP5-3.4over input datasets for CMIP were conducted by the REMIND-MAGPIE IAM, but experiments were also mirrored in other IAMs. Figure 3a illustrates that the IAMs are more in agreement on the carbon content of current total harvest, but they differ in future projections under the SSP5-3.4-over scenario.  Only a small subset of ESMs both conducted this simulation in CMIP6 and report harvest rates, but they are in significant disagreement about the current harvest level – highlighting a potential bias which would require further calibration if BECCS fluxes were calculated internally in ESMs.
We can get some intuition for the ESM simulated additional bioenergy production required for the BECCS-based carbon removal in SSP5-3.4over by assessing the difference between total harvest in SSP5-8.5 and the SSP5-3.4 overshoot (Figure 3b)
The difference in harvest in REMIND-MAGPIE  notably exceeds the difference between ESM simulated harvest flux in SSP5-85 (where there is no deployed BECCS) and SSP5-34-over in all 3 of the models considered (Fig. 3), indicating that none of these models would be able to replicate the level of negative emissions assumed in REMIND-MAGPIE – despite being driven by land use transitions derived from that model, potentially due to less productivity in ESMs in the areas designated for harvest..  Notably, other IAMs also vary significantly in their assumed harvest fluxes (indicating a varying reliance on BECCS for carbon capture).  Again, this highlights that if future climate simulations allowed BECCS fluxes to be calculated internally within the ESMs, there could be significant additional variance in the simulated forcing trajectory of any scenarios with substantial BECCS.."

[Figure]

[Figure]

[Figure]

Figure 3: (a) An illustration of total harvest carbon flux as simulated in the SSP5-34-overshoot scenario as simulated by the the SSP5 marker model (REMIND-MAGPIE, solid black) and other integrated assessment models (dotted and dashed black lines), compared with estimates from 3 Earth System models (colored lines)

which completed both simulations.  (b) colored lines show the simulated difference in ESMs between harvest carbon flux in SSP5-34-overshoot and SSP5-85.  Black lines show differences in total harvest in IAMs.

*RC3: Clarification of the ultimate purpose(s). In general, the manuscript does a fantastic job of highlighting a multitude of benefits of moving to an carbon-emission-driven framework. Maybe the authors could consider of systematizing some of their thoughts with regard to the ultimate purpose(s) – possibly contradictory – that can (or cannot) be better achieved with this change. From reading the manuscript, I can see many listed benefits, but am not 100% clear about the author's view on the ultimate purpose of an enhanced scenario design. I can extract for example the three following:*

1. *Creating more policy-relevant ESM runs.*
2. *Pushing the ESM development*
3. *Better Earth system understanding (e.g. land-based CDR potentials)*

*It might be worthwhile to flesh those (or similar ones) out more clearly and then discuss the listed benefits (or limitations) with respect to those ultimate purposes. This is just a suggestion that the authors might want to pick up.*

Thanks for this - we agree, and now frame the abstract and conclusion with the following key benefits of e-driven scenarios:

"These developments will allow three primary benefits: (1) resources to be allocated to policy-relevant climate projections and better real-time information related to the detectability and verification of emissions reductions and their relationship to expected near-term climate impacts (2) scenario modeling of the range of possible future climate states including Earth system processes and feedbacks which are increasingly well-represented in ESMs and (3) optimal utilization of the strengths of ESMs in the wider context of climate modeling infrastructure (which includes simple climate models and km-scale models)."

The third point is connected to RC2 - that increasingly the strength of ESMs is not in producing distributions of probable future climate, but rather to provide process resolved end-to-end case studies or point samples in a high dimensional process space - allowing for interactions between components, where coupled system effects can be assessed.

This partly reflects the strengths of ESMs in an increasingly diverse model space.  The low computational cost and small parameter space of simple climate models make them well suited to making probabilistic inference given observed global mean trajectories.  High resolution, atmospheric only models are well suited to assessing fine scale impacts and resolution limited processes.  But ESMs excel at coupled process representation to represent coupled system feedback processes which cannot be represented by either simpler models, or indeed high resolution models.

*RC4: More precision in terminology. I think the manuscript would greatly benefit from a more precise use of terminology, in particular with regard to the following three areas.*

1. *Scenarios generations. The authors seem to use SSPX-RCPY interchangeably with SSPx-y. For example, in Line 195, it says "The ScenarioMIP ensemble SSP5-RCP8.5". I would suggest to follow Table 1.4 in IPCC AR6 WG1.*

Good point.  Now using SSPx-y notation throughout

2. *CDR, land-based CDR, DACCS etc. Some statements seem imprecise as they refer to CDR in general, but discuss limitations that are only applicable to, for example land-based CDR, not DACCS or enhance weathering etc.*

Revised as suggested

3. *Emission-driven, activity-driven and concentration-driven. (Line 359, 361ff and others) In relation to the core issue of the manuscript. If I understand the authors correctly, then what is actually proposed are ESM runs that are:*
   1. *Emission-driven for fossil and industrial CO2 emissions.*
   2. *Activity-driven for land-use CO2 emissions.*
   3. *And concentration-driven (for the time-bring) for all other well-mixed GHGs.*

*Of course, saying 'fossil-industrial-CO2-emission-driven, land-use CO2 activity-driven, and other GHG concentration-driven' is a bit clumsy, so maybe the authors want to create an acronym for that? FIELAOC? (Am sure you can come up with better ones …). Alternatively, in line 261, you refer to it as an 'hybrid' approach. That might also be a good solution to consistently use throughout after defining it. While simply saying 'emission-driven' is nice and simple and useful for the high-level communication in a talk or press release, I think for many readers it can prolong confusion (congratulations on Figure 7, which nicely clarifies the approach). Maybe provide a table with those different 'hybrid' approaches?*

Thanks for this - we're now defining and using 'hybrid emissions driven' and using this term throughout the text

*RC5: Odd use of R-value statistics for TCRE and TCR in Figure 9 and text. The analysis presented in Figure 9a and b shows two metrics (TCRE and TCR) derived under the two different experiments (flat10 and 1pctCO2). Both in the Figure panels and the text, the high R-value is taken to argue that results from both experiments are interchangeably close (if I read correctly). Suppose that one experiment would result in exactly halved TCRE and TCR*

*values. The R-value could even be 1, but the experiments are suggesting very different TCRE values. Thus, the R-value is not the correct statistics here to make the point. A simple residual plot with the percentage difference between the two TCRE or TCR values would probably be much better suited to argue that differences are a) understandable and b) generally small.*

We are cutting this section from this paper, given the situation has moved on somewhat since the initial paper.  Flat10 is now part of the CMIP7 fast track, so we will publish the flat10 design and trial MIP separately.

*RC6: Better anchor section 4.1 on computational needs of high-resolution modelling. When reading the section, it felt a bit detached. There is a lot of great material in there, but I was not sure (and that might be just me) of how to relate the discussions of convection-resolving / convection-permitting ESMs and their computational needs to the question of the computational burden that interactive gas cycles add to the ESMs. I might have overlooked it, but if the authors could provide the reader with a clear indication of how much additional computational burden is created by integrating (a) the carbon cycle, (b) the methane cycle, (c) the nitrogen cycle and (d) any additional tracer to represent other well-mixed GHGs (another 40 of them are currently summarized in 'equivalent' concentrations), then that would better link this section to the rest of the paper IMHO.*

Thanks for this - we agree that this section was a little out of place, and we're reserving the plot for a future paper.  We have eliminated the section, and put in a paragraph on higher resolution modeling in the context of new section 4.4 (the 'challenge' of increased computational demand for e-driven runs.)

*RC7: Sketch the advance that is needed to diagnose land-use related emissions in ESMs. The paper touches this issue (e.g. Line 117ff), but I think it deserves a new community push (in particular as you have the world experts together in the author list). How can CMIP7 assist the ESM community to provide standardized diagnostics that tackle the intrinsically difficult issue of separating direct anthropogenic emissions from indirect anthropogenic emissions and natural fluxes? For runs that are communicated as 'emission-driven' (although part of it will be 'activity-driven'), it is still vital for readers and policy-makers to know what the emissions actually are. Correct me, if I am wrong, but if the diagnostic capability stays the same as it is now (in simplified terms: We do not really know how much additional anthropogenic emissions are created by an ESM due to a certain land-use activity and change of land-use patterns – partly due to a lack of consistent diagnostics across ESMs), then we either a) have to think about future workarounds to determine or approximate*

*landuse CO2 direct anthropogenic emissions or b) continue to fly blind in the sense that our headline results will be: "The remaining carbon budget for 1.7C is X GtCO2 fossil fuels and industrial emissions plus landuse activity patterns Y." (which would not be very catchy to say the least). Anyway, the point I am trying to get at: a clear recipe and renewed community push for either good workarounds or other stabs at this seemingly intractable problem of diagnosing landuse related emissions from ESMs would be great. And I am not saying that this problem is created by shifting to activity-driven runs, but it is drawn onto the stage.*

This is  good point - we've added a section to address:

"### 1.1.1    Diagnosis of land use emissions

There remains significant uncertainty in both the simulation and the assessment of observed emissions due to land use change (P. Friedlingstein et al. 2022). In concentration-driven simulations in CMIP6, land use emissions calculated internally in each model, and were consequential in terms of derived compatible fossil emissions (Liddicoat et al. 2021), and land use emissions are assessed independently in LUMIP (Lawrence et al. 2016).   However, there remains significant uncertainty on the definition and quantification of land use fluxes.   In the Global Carbon Budget(P. Friedlingstein et al. 2022), for example, best estimates of land use emissions are derived from bookkeeping models (Hansis, Davis, and Pongratz 2015; Houghton and Nassikas 2017; Quilcaille et al. 2022) which use empirical growth curves to estimate the transient carbon stock response to land use changes. Meanwhile, national inventories use different accounting conventions to those used in IAMS, ESMs and bookkeeping models – including not just transitions in land use, but also including land sinks in some regions whose usage remains static, but which are designated as managed (Gidden et al. 2023).

And in 5.5.2:

A hybrid emissions-driven design would place heightened importance on the ESM calculated land use fluxes, which would influence downstream climate impacts directly.  However, in a full transient historical or future simulation, it would be difficult to directly isolate the fraction of net land-atmosphere carbon exchange which  is associated with land use change and the fraction associated with natural carbon sinks evolving over time under changing climate background states (notably, this is a challenge in assessment of national inventories also (Gidden, 2023)).  As such, additional diagnostic counterfactual experiments such as those provided in LUMIP are essential. In CMIP6, these experiments were limited to a concentration-driven framework (e.g. LUMIP experiment *hist-noLu,* a variant of the concentration-driven historical simulation with no land use change).

In the hybrid emissions-driven model, such diagnostic experiments need to be expanded to include emissions-driven experiments to capture the contribution of land use changes to net transient land use fluxes in the coupled simulation.  An *esm-hist-noLU,* for example, which followed the protocol of *esm-historical* with fixed land use change, would differ from esm-historical both in terms of the effective land use emissions, but also in terms of any ensuing carbon-climate feedbacks which could

modulate the natural emissions also, resulting in counterfactual historical carbon dioxide concentrations..  As such, a full understanding of the role of land use in the transient land sinks in emissions-driven simulations will require a carefully designed set of complementary diagnostic experiments for both historical and future simulations, likely including both emissions-driven and concentration driven diagnostic experiments (for example, with prescribed concentrations from the fully coupled simulation)..”

*RC8: Figure 5. There are two suggestions for Figure 5.*

> *Add extra panel with extension to 2100 under esm-ssp5-8.5. For example, similar to Figure 2a in Friedlingstein et al., 2014. (DOI: 10.1175/JCLI-D-12-00579.1). The reason is that back then, the IAM / MAGICC combination tended to be towards the lower end of the range of projected CO2 concentrations. With an adjustment of the carbon cycle dynamics towards the representation of this CMIP5 range of carbon cycle feedbacks, the plot until 2100 should be expected to show the opposite of Figure 2a in Friedlingstein et al. 2014, i.e. that overall the CMIP6 emission driven runs now show lower future CO2 concentrations for the same emission pathway compared to CMIP5 (or a CMIP5 calibrated emulator) (all other things being equal).*
> *More importantly though: The lower plots b and c are used in the text as justification that the additional uncertainty of 40ppm of historical mismatch of CO2 concentrations does not matter for temperatures. That is a somewhat surprising statement, as it equates to roughly 20 years of climate change trends (of course, there are other radiative forcers influencing that trend as well.). Nevertheless, a difference is indeed hard to decipher by eye in a spaghetti plot with thick lines, so I would ask for an alternative or additional visual, which is a panel with histograms of last 20-years of tas in esm-historical and historical runs, with the two histograms plotted on top of each other plus their relevant medians and ranges ... Even more ideal would be a scatter plot that indicates last-20year averages of tas on the x-axis, the last-20years of CO2 concentrations on the other and then having one cloud of dots for 'esm-historical' and one assembly of dots (on the same CO2 level horizontal line) for 'historical'. In that way (and in particular, if the marginal distributions are shown also by histograms), the challenges that can potentially arise from shifting to emission-driven runs could be better anticipated and discussed.*

[Figure]

Many thanks for these suggestions - in fact, we revise our assessment of this plot in light of your pointers. We add boxplots for warming in the 2005-2015 period for esm-historical and historical, using 2 baselines (1850-1900) as before, plus (1970-1990) to resolve recent warming trends. In both cases - the inter-quartile range is notably wider for esm-historical than for historical. Note - we are not including future information in this plot because the section refers explicitly to the difficulties of historical calibration of e-driven models.

Revised paragraph:

"A challenge with running models in hybrid emissions-driven mode is the additional degrees of freedom associated with calibrating the coupled climate-carbon cycle system to reproduce both the joint evolution in historical concentrations of climate forcers and the historical warming increases. CMIP6 esm-historical simulations show most models (10 out of 13 models in C4MIP) fall within a range of $CO_2$ concentration range of 40ppm – representing some 20 years of historical emissions. This is significantly greater than the observational uncertainty (about 0.1ppm(Pierre Friedlingstein et al. 2022; Lee et al., n.d.)), and results in some increase in the model uncertainty in warming represented by the distribution of historical warming in CMIP6 simulations and their concentration-driven historical analogs for models which completed both experiments (Figure 5b,c) – and inter-quartile range of 0.45K for warming in 2005-2014 compared with an 1850-1900 baseline in esm-historical, compared with 0.25K in the concentration driven historical experiment. Notably, using a more recent baseline period (1970-1990), the 'hot model' issue of overestimated recent warming (Hausfather et al. 2022) is apparent by considering the concentration driven historical recent distribution in the context of observations (figure 5d), but the higher variance of recent warming in the emissions-driven simulations result in the observed warming lying within the inter-quartile range of simulated warming."

*Comments on specific lines:*

> *Line 60ff: All the references need a good second look for appropriate formatting.*

Thanks, will revise

> *Line 67: "In CMIP6, scenarios were … " See RC4a) above. This is confusing. Make consistent with terminology in IPCC AR6 WG1 Table 1.4.*

Updated as follows:

" In ScenarioMIP/CMIP6, scenarios were in terms of SSPs which defined broad socioeconomic background states, where which constrained the global mean end-of-century radiative forcing targets (O'Neill et al. 2016; Riahi et al. 2017). IPCC AR6 ("Framing, Context, and Methods" 2023) adopted the notation of SSPX-Y, where X is one of 5 SSPs, and Y is the radiative forcing level used in the creation of scenarios for ScenarioMIP"

> *Line 78: For the CMIP pipeline, add the relevant reference of Kikstra et al. (https://doi.org/10.5194/gmd-15-9075-2022) and Gidden et al. (https://gmd.copernicus.org/articles/12/1443/2019/).*

Thanks, done.

> *Line 90: Rather than the Technical Summary, Cross-Chapter Box 7.1 in IPCC AR6 WG1 is probably what is most appropriate here?*

Thanks, added as suggested.

> *Line 91: Very nice graph. You could refer to similar earlier figures for context for the reader, such as Figure 1 in Cross-Chapter Box 11.1 in IPCC AR6 WG1. Just an option.*

Thanks - added the following:

"In some cases, climate assessments bypass the causality chain and express impacts as a function of global mean temperatures (Figure 1 and cross-chapter box 11.1 in ("The Earth's Energy Budget, Climate Feedbacks and Climate Sensitivity" 2023)].  )"

> *Line 119: Explain what you mean by "these differences are counterintuitively manifested in compatible emissions… "*

Deleted the word "counterintuitively"

> *Line 123, Figure 2: Check the figure panel numbering, a,a,a,c,d,d*

Fixed, thanks

> *Line 123, Figure 2: The Figure legend says GCP2022, and the reference in the caption is provided with P. Friedlingstein. Make consistent.*

Done

> *Line 123, Figure 2: The dotted line says "IAM". Now, the emissions are from each of the IAMs, but the comparability to the ESM lines is contingent about the SCM/MAGICC step that was used to provide the concentrations from which the ESM inverse emissions are derived. You might want to unpack this.*

Good point - added reference to Meinshausen 2020.

> *Line 123, Figure 2: The difference in historical emissions from 1950 to 1990 is rather large between GCB2022 emissions and most of the ESM inverted emissions. I think a paragraph on this would be worthwhile (see also RC1.1 above).*

Added a brief discussion on historical bias, following the future discussion.

> *Line 131ff: "simulated net-zero dates measured in terms of compatible… ". This sentence seems to suggest that the IAM net-zero date is biased towards early dates. Maybe add clarity here in terms of what your thoughts are: Is that a systematic over-estimation of carbon cycle feedbacks by the IAM/ MAGICC combo? Or are the recent ten years of observations, in which the ESMs also over-estimate anthropogenic emissions for the observed atmospheric concentrations (see peak of the trajectories around 10 GtC in Figure 2 panel SSP1-2.6). indicate a potential issue elsewhere?*

It's an interesting point to follow up - and we don't want to speculate here as to the correct interpretation.  We've highlighted the issue as follows:

".  The fact that the IAM/MAGICC estimate lies on the edge of the ESM compatible emissions distribution is worthy of further consideration, either indicating that MAGICC carbon-climate dynamics are a slight outlier amongst the ESMs, or a methodological difference between the compatible emissions in the ESMs and the harmonized emissions trajectory produced in the IAM/MAGICC pipeline (Meinshausen et al. 2020) .  "

*Line 140: As you mention other SCMs by name in the manuscript, maybe here as well ?:-)*

MAGICC gets plenty of mentions in the revised version :)

*Line 163-164: "while this approach is convenient, bypassing process-resolving … has risks". I'd suggest a bit more nuance here. If a simple climate model with an uncalibrated carbon cycle is used 'in the assessment of mitigation strategies', then I fully agree. However, if an emulator that is properly calibrated to exactly capture the process-resolving carbon cycle uncertainties from ESMs, then an emulator could be a tool to assist translating the limited number of ESM process resolving carbon cycle uncertainties for policy-relevant settings. Or how would the authors suggest the >1000 scenarios assessed in WG3 should be evaluated in terms of their concentration and temperature outcomes. One option is to use properly calibrated emulators / SCMs. Anyway, the authors may want to add nuance to that sentence.*

Agree on this - and we've refined the argument a bit as follows:

"Simple climate models are well suited to this application – with sufficient structural complexity to emulate more complex models, but sufficiently computationally lightweight to allow rapid sampling of a relatively low parameter space to find model variants which are consistent with observations (C. Smith et al. 2024; M. Meinshausen, Wigley, and Raper 2011).  The increasing use of simple climate models in assessment (Nicholls et al. 2022) as the primary mechanism for representing uncertainty in global scale climate response allows Earth System Model simulations in CMIP to focus on coupled complex process representation.  A CMIP ensemble with a primary focus on emissions-driven scenarios, starting with $CO_2$ emissions in CMIP7 but with a longer term objective to represent human activity through diverse emissions or land management, would allow ESM scenarios to represent real-world climate policy and its outcomes.  As emissions and activity-driven processes are improved in ESMs, it is essential that SCMs can emulate any new emergent global coupled dynamics which arise in the ESMs (e.g. nonlinear behavior or tipping points).  In short, the presence of a larger model ecosystem including ESMs, SCMs and km-scale models allows for each model class to excel in dimensions which are suited to the platform.  For ESMs, the computational efficiency and resolution must balance the need to represent coupled complex processes with the need to be able to calibrate and spinup the coupled system.
"

*Line 183: "The plausibility of large scale CDR.." here and elsewhere: make terminology precise – unless you refer to all CDR (see RC4.b above).*

Clarified (we do mean all CDR here)

*Line 184: "Interventions will cause ….". Well, probably you refer here to land-based CDR, rather than DACCS or enhanced weathering etc..? Please rephrase/clarify.*

Done.

*Line 195: "ScenarioMIP ensemble SSP5-RCP8.5". See comment RC4.a above.*

Now using SSPX-Y notation throughout.

> *Line 202: "about available CDR capacity"... see comment RC4.b above. This comparison to crop harvest probably has little relevance for DACCS capacity. Please clarify, e.g. by simply saying 'land-based CDR' or similar. See also RC2 above.*

Rewritten

> *Line 212-214: "Ocean carbon balance is also ... (Navarra and Fogli, 2013)." The uninitiated reader might think that this is a particular limitation of land-based CDR. Without additional explanation, I think such statements can be misleading, as they do not invalidate the basic principle that to offset one tonne of emissions you need to have one tonne of permanent removals. The dynamics of recalibrating carbon pools are scientifically interesting but can be misunderstood – as in this non-contextualized sentence. Please clarify.*

Deleting this - agree with the potential confusion.

> *Line 220: Change "where SCMs and emulators..." to maybe "which also results in SCMs and emulators... ". You are the native speakers, though.*

Changed to "cases in which SCMs and emulators…"

> *Line 225: If the "dynamics between carbon removal and the wider climate state" refer to the sentence in line 212, then I am not quite sure about this statement. Maybe clarify what exactly you refer to?*

Deleted this sentence.

> *Line 226: "As such, concentration-driven ... ". Is that an issue with IAMs or with the concentration-driven setup? I would argue it is the former. Certainly, an emission-driven or activity-driven setup within ESMs helps, as thereby this IAM limitation is circumvented and ESMs are 'forced' to implement the land-use activities in their framework, but whether ESMs then get emission-driven or concentration-driven should not matter to the outcome (as appropriately derived inverse emissions and activity options within an ESM for a prescribed concentration-driven setup could also work, albeit being slightly complicate). No? Please clarify.*

Added the following clarification:

" An emissions-driven framework would directly assess these risks associated with land-based carbon mitigation (such as through afforestation, reforestation, forest management, biochar, agricultural soils or BECCS), by providing a range of potential outcomes for the land and ocean-based removal strategies which are employed in the scenario which can contextualize and provide uncertainty bounds for the climate trajectory simulated internally within the IAM."

*Line 229: Referring to RC4c above, it would probably be more appropriate to talk here about an "activity-driven" or "pattern-driven" etc. approach, rather than emission-driven?*

Agreed - now using 'activity driven' in this section.

*Line 243 (see also line 508): While I fully agree that the 'radiative forcing' label is well overdue to be replaced by an emission-based metric, such as 'cumulative CO2 emissions' or similar, the manuscript could be more nuanced here, clarifying that these radiative forcing labels are of course only indicative/illustrative labels. The radiative forcing within each ESM for the same scenario wildly differs.*

We've clarified this point.

*But it would be good to hear the authors suggestions of how the next generation of scenarios should be labelled. I see the suggestion in line 508 of "should be categorized in terms of their policy strategies (e.g. net-zero dates and land use strategy) but how exactly do that in practice? (and I am sure than all of these suggestions are better than the current RF label ).*

Added some more thoughts on this:
" Furthermore – the ability to simulate different types of carbon removal processes and non-CO2 mitigation strategy within the ESM opens the door to having multiple scenarios with comparable best estimate temperature outcomes in the IAM, but with different uncertainty ranges simulated in the ESM ensemble. As such, the naming strategy for emissions-driven scenarios will ultimately need to represent a higher dimensional space, providing a shorthand for embedded characteristics on decarbonization rate, removal strategy and nonCO2 emissions. This may be more easily achieved with qualitative identifiers than with continuous labels referring to radiative forcing or temperature targets."

*Line 261ff.: It took the Introduction almost 10 pages to reveal to the reader a bit more detail of what you are actually proposing – and very useful additional detail is provided later on in the manuscript. Consider shifting a clear succinct paragraph of your proposal further up in the Introduction.*

Agreed - we've shifted this section into the end of the introduction 1.1.

*Line 310: Well, the issue is a bit graver than presented here. The issue, as highlighted by Grassi et al. and others, is that countries are currently allowed under the emission inventory guidelines to account for natural responses of the carbon cycle (aka 'CO2 fertilization' in forests that the countries declare as 'managed') to count as removals. Thus, UNFCCC LULUCF emission submissions contain not only 'directly induced anthropogenic emissions', but also a part of the*

*natural carbon cycle response. Anyway, see the Allen et al. piece on geological net-zero under preparation (would be good, if timing works out that you can refer to it) to point to the real problem here.. the non-permanence of sinks is one important issue. But the fact that countries count something that is not actually directly anthropogenic is an even bigger issue (as by that logic, one could argue that the non-airborne fraction of CO2 should be counted towards emission inventories, which would then allow countries to be net-zero, while in fact we only halved anthropogenic emissions). Anyway, I would strongly suggest to add this important issue as ESMs with an interactive carbon cycle and realistic land-use patterns could indeed greatly help to separate out direct anthropogenic from indirect anthropogenic (e.g. CO2 fertilization) fluxes and thereby help changing a major shortcoming in IPCC methods for UNFCCC reporting guidelines.*

Fully agreed - we now mention the Grassi issue:

"Meanwhile, national inventories use different accounting conventions to those used in IAMS, ESMs and bookkeeping models – including not just transitions in land use, but also including land sinks in some regions whose usage remains static, but which are designated as managed (Gidden et al. 2023; Grassi et al. 2021).

Better land use process representation in an emissions-driven framework, must therefore be supported by diagnostic simulations to map between these accounting systems…."

*Line 326f: .. but none provide a fully self-consistent internally generated representation of the chain of causality from emissions to concentrations", while this is an excellent phrasing of one of the purposes that such ESM scenario design should pursue, maybe there are similarly important 'purposes' that either align with this objective or conflict with it (given overall resource constraints). I think of the objective of attempting to end up with a comprehensive fully probabilistic representation of our current knowledge across each of the cause-effect chain links.. that can be addressed by ex-post analysis and emulators, but the mere 'self-consistent internally generated representation' cannot be the only ultimate goal, if we end up with just a sparse sample of five-star ESM realisations. Anyway, maybe consider rephrasing. See comment RC1.2 and RC1.3 above.*

Reworded to be more complementary:

"These estimates would be well supported by fully self-consistent internally generated representations of the chain of causality from emissions to concentrations which could be achieved in emissions-driven ESM simulations. "

*Line 366ff: A table would be a great resource that summarizes CMIP6 model capabilities as of now. Also, this table could contain a comparison of the*

*emission-driven concentration hindcasts with observed concentrations as shown in Figure 5, say for the last 20-year averages...*

The table exists in the reference! (Seferian 2020)

*Line 388: See comment RC8 above.*

*Line 388: One could add the reference for the historical CO2 concentrations used in CMIP6 into the list of references at the end ...*

Done

*Line 395-400: The section "Although this is ... biases in CMIP6 (Papalexiou et al. 2020)" seems to suggest that it is already established that 40ppm CO2 differences are 'not a significant factor' in historical warming in esm-historical runs. See my comment RC8 (2) above. Maybe I misunderstand, but I do not think that this conclusion can be robustly drawn from the two spaghetti plots in Figure 5, nor does Papalexiou et al. 2020 seem to consider the esm-historical emulations, but rather focusses on the concentration-driven results (if I am not mistaken). Thus, please consider adding more evidence to this section and your conclusion that 40ppm CO2 differences do not matter. If they turn out to matter, please discuss potential solutions (e.g. would a hybrid protocol, where historical runs are performed concentration driven, but future runs would be emission/activity driven work? Why or why not?).*

See answer to RC8 - we've revised this position following your suggestions.

*Line 415: When you say "it is equally important to sample inter-model uncertainties ... as for the physical climate response to a single trajectory of CO2"... Please clarify. Do you mean that CMIP7 should contain one emission and one concentration-driven variant of each scenario?*

Simplified sentence: "As such, we argue that in order to provide robust information for both adaptation and mitigation, it is equally important to sample inter-model uncertainties in the wider carbon-climate system. "

*Line 435ff: Very nice set of suggestions. Please consider whether you can expand with the potential elements of a protocol that would allow the robust delineation between directly anthropogenic, indirectly anthropogenic and natural carbon fluxes (see comment RC7 and on line 310 above).*

Added the following:

 "Increasing understanding of how to map between national accounting systems and ESM/IAM output (Gidden et al. 2023; Grassi et al. 2021) can be strengthened with hybrid emissions-driven simulations (combined with well chosen counterfactual experiments in LUMIP), where ensembles can provide ranges of modelled direct and indirect anthropogenic fluxes from land use change."

*Line 461: Not all readers might be familiar with the term "digital twins". Please explain.*

Added a citation (https://www.nature.com/articles/s43017-023-00409-w)

*Line 493, Figure 6: See comment RC6 above. Please consider how to visually integrate the computational burden for carbon, methane and Nitrogen cycles into this graph, which otherwise feels a bit detached from the rest of the paper.*

Agreed - we've removed the graph.

*Line 507: What do you mean by 'original IAM simulation'? Do you refer to the temperatures assumed within, e.g., IMAGE, or the temperatures that are produced by the 'harmonising' SCM MAGICC to produce all temperatures for all the SSP scenarios (and associated concentrations?).*

The latter, clarified as follows:

"(currently harmonised SCM simulations combining historical climatic trends and IAM driver data) "

*Line 523, Figure 7: Excellent figure. Two remarks:*

*In panel c: If the land-based CDR activities are already part of the top arrow "Land transitions (impacting concentrations)", then why are "CDR emissions" listed separately in the lower-right arrow description "Fossil CO2 & CDR emissions"?*
*Should the latter arrow rather read "Fossil/industrial CO2 emissions and non-land based CDR", given that land-based CDR is already part of the land cover information?*

*In panel d: What is the separation between the "Land transitions" arrow at the top right and the "CDR" part in the "CDR & SRM strategy" part in the lower right?*
*Should the lower right again be called "non-land-based CDR & SRM strategy".. and rather than "strategy", should it be called "activities" (as ESMs are unlikely to include a policy module to estimate a level of activity change from a 'strategy'?!)*
*In panel d: The lower right arrow "all emissions". Should that rather be called "Fossil/industrial CO2, Fossil/industrial CH4 and Fossil/industrial N2O emissions" given that land-use emissions will be generally estimated from within the ESMs based on activity data provided by IAMs?*
*In panel d: If there is no arrow any more for some non-CO2 concentrations, what is the suggestion what should happen with the 40 smaller GHGs, i.e. the CFCs, HFCs,*

*PFCs, HCFCs, Halons etc? Is the suggestion that the ESM will have tracers and gas cycles for each of these GHGs? Probably not, I would assume. So some small box for an offline SCM producing those concentrations and equivalent concentrations in the form of 1 or 2 or 3 'equivalent' tracers will probably still be necessary in CMIP8, right?*

Updated as suggested (apart from the very last point - which we clarify in the caption)

[Figure]

(a) GHG concentration driven model, predefined aerosols ("CMIP3")

[Figure]

(b) GHG concentration driven model, interactive aerosols ("CMIP5,6")

[Figure]

(c) Partial GHG e-driven model, interactive aerosols ("CMIP7")

[Figure]

(d) Complete e-driven model, interactive aerosols, SRM & CDR ("CMIP8+")

[Figure]

*Line 525, Figure 7 caption: When stating "a proposed CMIP8 pipeline, with complete emission driven configuration". Please consider whether you want to be more precise (see comment RC4c and comments on line 523 above. I would assume that you do not suggest to provide 40 emission trajectories of fluorinated gases to each ESM, so 'complete emissions driven' is maybe not the appropriate term. It comes back to whether you might want to define those (somewhat complex) hybrid running modes in a table with acronyms or shortcuts, so that you can provide the details once and then refer to here ane elsewhere.*

Updated caption:
"Figure 7: Stylized illustrations of the historical (a,b) and proposed (c,d) information flow for CMIP. (a) shows concentration-driven modeling pipeline with prescribed aerosols common in CMIP3 (b) shows concentration-driven modeling pipeline with interactive aerosols common in CMIP5,6 (c) a proposed scenario pipeline for hybrid emissions driven simulations in CMIP7 with carbon emissions but maintaining concentration definitions for non-CO2 greenhouse gases (d) a proposed CMIP8 pipeline, with emissions driven configuration for CO2, N2O and CH4 and process based implementation of CDR and SRM approaches
"

*Line 532: Is "to represent" the correct verb here?*

Changed to "process"

*Line 551, section 4.4.1.: To me, the closed cycles for water and for other GHGs are separate issues which might warrant distinct subsections. Please consider.*

Point taken - but in the interests of not further inflating a long paper, we will keep as is.

*Line 560: Again, what happens to the 40+ smaller GHGs?*

Added to the main text:
"As such - pragmatically, on a timescale of CMIP7, there will remain elements of historical and future simulations which will, for most models, remain exogenously defined but developments could be considered for CMIP8 and beyond (Figure 7d), though it remains likely that some concentration-driven elements will persist – given the large number of minor climate forcers currently handled by SCMs (CFCs, HFCs, PFCs, HCFCs, Halons etc) (Malte Meinshausen et al. 2020)."

*Line 571: Something missing after "... found in "?*

Deleted

*Line 573: Regarding the text ".. are at least representative of those obtained from ESMs". Now the question is are those land-use fluxes available from ESMs? So far, there is great diagnostic difficulty of estimating the land-use based anthropogenic emissions, so am not sure I understand the sentence of what it means in practice. Please clarify.*

Deleted.

*Line 602, Table 1. Consider separating the column "forcing" into "CO2 forcing" and "non-CO2 forcing" for clarity.*

Done

*Line 602, Table 1: Should "Metrics provided from simulation" rather read "Metrics derived from simulation"?*

Deleted

*Line 602, Table 1: When it says "Linearly declining emissions by 2 GtC/decade from 10GtC/yr (year 100) to -10 GtC/Yr (year200)...". Please check the units. I think you can either write "... by 2 GtC/decade from 10GtC to -10GtC...", i.e. leaving out the "per year". Or you the first one will need to be "by 2GtC/yr/decade" as you refer to a rate of change to the rate of annual emissions, if I understand correctly.*

Good point, updated.

*Line 602, Table 1: For 'esm-historical' last row and fourth column 'Forcing" it says "historical emissions". I presume you mean "historical fossil & industrial CO2 emissions plus land-based activities" or sth similar to that? (and again, please specify for non-CO2 in a separate column for clarity).*

Yes, Updated as you suggest

*Line 617: See comment RC5 above.*

This plot is deleted now (will be in a dedicated flat10 paper)

*Line 628, Figure 9: see comment RC5 above.*

This plot is deleted now (will be in a dedicated flat10 paper)

*Line 631: When you list "emissions are constant throughout.. " as one of the "desirable properties". It seems more policy-relevant and more desirable than the emission profile under 1pctCO2, but an even more desirable property would be a 'gradually declining to zero' shape, which better resembles policy-relevant settings, no? Consider rephrasing.*

We cover this in flat10-cdr (and a potential variant flat10-nz, where emissions remain at net zero after a linear decline).

*Line 638: On the statement "we find both a generally slightly higher... " – yes, I think that is the actual comparison metric (rather than the R-squared statistic). Please expand on how much higher and why (I think you have the material further below, but not the quantitative numbers, which are important in order to put any CMIP7 TCRE results into context of CMIP6 TCRE results.)*

Deleted this also, for the flat10 writeup

*Line 649, Figure 10. The argument that the correlation in the panel c plot is a closer one compared to (a) and (b) is not fully convincing due to the vastly different scales of the plots. Consider plotting the residuals towards the diagonal instead (or in addition), which would more clearly show whether a 'higher TCRE' is partially offsetting a 'lower ZEC', or similar. A scatter plot (similar to the one I outline in RC8 (2)) would be even more ideal to make the point.*

Deleted this section.

*Line 657: On the 'around 10%'. From the plot, it looks that the differences for the higher ZECs are substantially higher than 10%. Is 10% the mean deviation? Maybe provide mean deviation and ranges. Please clarify.*

Deleted this section.

*Line 659: By logical conclusion, a TCRE that is evaluated at the point of zero emissions after 75 years of 10 GtC emissions and then a 50 year ramp-down period to zero would likely create an even better approximation of peak warming (i.e. ZEC after such a scenario would be even smaller?!). I note the nice indicator TNZ that basically addresses this slight remaining shortcoming of deriving TCRE from esm-flat10. Please clarify why you do not propose that design for the TCRE derivation (which would make the TNZ 'bias' metric superfluous, it seems).*

Deleted this section.

*Line 703: Excellent set of indicators, very useful. One can already see the nice papers emerging from those diagnostics. Consider putting into a table instead and consider also to 'explain' the abbreviations 'tPW', 'TNZ', does TNZ stand for 'Temperature bias Net Zero'? Probably not .. .*

Deleted this section.

*Line 740ff: When you say "The ECS could be inferred indirectly by calibrating a simple climate model to the ESM output of esm-CDR-pi-pulse… " it send shivers down my spine – The precise calibration with a simple SCMs can introduce so many uncertainties when using only an single idealized scenario, that I really doubt the robustness of the derived ECS and the comparability to ECS derived from long-term 2xCO2 or 4xCO2 simulations. Also, how would be characterise time-changing or state-dependent feedbacks over time (Sheerwood et al., https://doi.org/10.1029/2019RG000678; Rugenstein et al., 2019, https://doi.org/10.1029/2019GL083898)? Please reconsider.*

Deleted this paragraph. We agree that the a concentration driven abrupt4x and 1pctCO2 remains a useful diagnostic.

*Line 758f: "A decade earlier… ". Yes. Very important sentence.*

*Line 766: After "with the right experimental design", it would be good if you cite the current community considerations on that design, namely the ScenarioMIP draft proposal (e.g. the Reading workshop document, unless the consultation document of ScenarioMIP is available in time), the much beloved () collection of thoughts in https://gmd.copernicus.org/preprints/gmd-2023-176/ (now accepted) and the ones by Pirani et al. 2024 (https://www.nature.com/articles/s44168-023-00082-1).*

Added the following:

"A draft scenario design document for ScenarioMIP CMIP7 indicates a request for a higher fraction of emissions-driven scenarios (Task Groups n.d.), and perspectives on the CMIP7 scenario design have called for higher relevance to Paris Agreement objectives through 'representative emissions pathways', exploration of CDR risks, and potentially counterfactual scenarios (Malte Meinshausen et al. 2023) while others have called for greater integration into the needs of multiple IPCC working groups and policy relevance (Pirani et al. 2024). Many of these issues can be addressed in a framework enabling an operational assessment of emissions-based policies. "

*Line 809, Appendix A1. The title just says "Additional idealised experiments NOT in the main ESM-DECK proposal". Please add some text as this is unclear. Do you suggest that these experiments (1) should NOT be considered as part of the ESM-DECK proposal, (2) are alternatives to the ones proposed in the main section of the text or are (3) complementary optional additions to the ones in the main text?*

The second option - now clarified. These are diagnostic simulations for comparison with the main recommendation.

*Tables A1 and A2: For clarity, again split the "Forcing" column into "CO2 forcing" and "Non-CO2 forcing".*

These are now deleted.

*Appendix B: It is not quite clear to me what the relevance of this table is for the paper – unless the paper provides more details as to how much computational demand arises from running ESMs with an integrated CO2/ methane / nitrogen cycle etc.. Please clarify.*

Deleted.

*And lastly: Deep apologies for the long review. I know how painful they can be. It was a very interesting read and is an excellent and important paper and I hope my review comments just make a strong paper stronger.*

No problem, many thanks for taking the time!

*First, I would like to extend my sincere apologies for the delay in reviewing the paper. The topic that the paper addresses, a call for (CO2) emissions-driven simulations for CMIP7 and beyond, is undeniably crucial. It is evident that this paper holds substantial importance and merits publication. I agree with the authors regarding the necessity of improving Earth system processes in ESMs, which is as critical as the pursuit of high-resolution km-scale climate model development. This aspect is often been overlooked recently, particularly as (too much?) attention tends to be directed in high-profile publications towards km-scale climate modeling. However, before I can consider the paper for publication, some comments require attention.*

Many thanks for the positive assessment.
* * *
**Response to reviewer 2**

*Major comments:*

1. *Prioritization of CO2 emission-driven simulations*

*While I appreciate the focus on CO2 emissions-driven simulations, I have some reservations about prioritizing them over concentration-driven simulations. As outlined in the manuscript, conducting simulations in CO2 emissions-driven mode introduces additional uncertainties, making it challenging to compare the physical climate, but also the biogeochemical response with observations over the historical period, mainly because the atmospheric CO2 concentration in these simulations does not align with observed levels. The authors seem to argue that a difference of about 40 ppm does not matter. However, this difference matters for historical ocean acidification, or any other variable that strongly depends on the atmospheric CO2 level. In addition, it becomes problematic to evaluate the models' ability to replicate, for example, decadal-scale hiatuses in global temperature, such as the one observed between 1998 and 2012, when changes in atmospheric CO2 concentrations and therefore the radiative forcings are already inaccurate. The same issue arises when assessing, for example, the recent anomalously warm year 2023. Given these*

*concerns, I believe concentration-driven scenarios remain essential and should be prioritized alongside CO2 emissions-driven runs.*

Many thanks for this point, and we have revised some of our arguments on the value of relying on e-driven simulations alone.  We now introduce a section 4 on the "limitations and new challenges" of the emissions-driven approach, including a longer section on coupled system biases - particularly associated with bias in present day $CO_2$ concentration.  We also include a more robust assessment of the spread in concentrations and temperatures in CMIP ESM experiments, and withdraw our conclusion that the additional variance is inconsequential.

The key section follows:

[Figure]

Figure 5: Carbon dioxide concentrations (a) and temperature anomalies (c,e) in emissions-driven historical simulations in CMIP6, and temperature anomalies concentration-driven historical simulations (b,d). Middle and bottom rows are relative to an 1850-1900 and 1970-1990 baseline respectively. Observed temperature data is from (Cowtan and Way 2014) Boxplots show the inter-quartile range (boxes) and 10-90 percentiles (whiskers) of the model simulated anomalies for the period 2005-2014.

A challenge with running models in hybrid emissions-driven mode is the additional degrees of freedom associated with calibrating the coupled climate-carbon cycle system to reproduce both the joint evolution in historical concentrations of climate forcers and the historical warming increases. CMIP6 esm-historical simulations show most models (10 out of 13 models in C4MIP) fall within a range of $CO_2$ concentration range of 40ppm – representing some 20 years of historical emissions. This is significantly greater than the observational uncertainty (about 0.1ppm(Pierre Friedlingstein et al. 2022; Lee et al., n.d.)), and results in some increase in the model uncertainty in warming represented by the distribution of historical warming in CMIP6 simulations and their concentration-driven historical analogs for models which completed both experiments (Figure 5b,c) – and inter-quartile range of 0.45K for warming in 2005-2014 compared with an

1850-1900 baseline in esm-historical, compared with 0.25K in the concentration driven historical experiment. Notably, using a more recent baseline period (1970-1990), the 'hot model' issue of overestimated recent warming (Hausfather et al. 2022) is apparent by considering the concentration driven historical recent distribution in the context of observations (figure 5d), but the higher variance of recent warming in the emissions-driven simulations result in the observed warming lying within the inter-quartile range of simulated warming.

Having only coupled simulations available would likely increase the difficulty of isolating the sources of bias in simulations (i.e. isolating biases in the ecosystem and physical systems). As such, fully coupled simulations would be well complemented by concentration-driven simulations if sufficient computational time is available to assess the role of coupled processes in model bias. Diagnostic offline land and ocean simulations to assess carbon cycle evolution under observed historical climate (Peng et al. 2021; van den Hurk et al. 2016). Such influence can go both ways: biases in the climate simulation can equally impact simulations of the ecosystem (Ahlström, Schurgers, and Smith 2017), creating differences in simulated carbon cycle responses between offline land simulations and fully coupled simulations. Bias correction approaches (Piani, Haerter, and Coppola 2010) should therefore be used with care in highly coupled simulations where biases in historical carbon or physical simulation can arise due to errors elsewhere in the model, necessitating additional caution when bias correcting model output for impact assessment (Maraun et al. 2017; Lafferty and Sriver 2023)

*1. CO2-emission driven simulations with fully coupled ESMs are not a new thing*

*After reviewing the manuscript, it appears that CO2 emissions-driven simulations are portrayed as a novel concept. However, it's important to note that earlier research has indeed utilized CO2 emissions-driven simulations. For instance, Joos et al. 1999 (https://www.science.org/doi/10.1126/science.284.5413.464) and subsequent works such as Fung et al. 2005 (https://www.pnas.org/doi/full/10.1073/pnas.0504949102)) have employed such simulations (the authors already cite the Cox et al. study). While the authors have partially addressed this concern in lines 360-368, it is worth noting that there are numerous additional studies beyond the CMIP realm that should be acknowledged. The authors should acknowledge more of these earlier studies and incorporate a paragraph addressing this in the manuscript. Otherwise, the paper may inadvertently give the impression of introducing a significant step change in Earth system modeling, when in fact, this approach has been utilized for some time but hasn't just been fully integrated into the CMIP exercise.*

Agreed - we now discuss these early studies in more detail.

"Past phases of CMIP have defaulted to concentration-driven scenarios, but models capable of running with a closed and interactive carbon cycle have been developed by some centers for over two decades (Cox et al. 2000; Joos et al. 1999; Fung et al. 2005), with intercomparison efforts for coupled carbon Earth System Models coming soon after (P. Friedlingstein et al. 2006; C. D. Jones 2020). These early studies established the significance of coupled carbon-climate processes in the wider evolution of the Earth System, with potential interactions between carbon balance and ocean

circulation (Joos et al. 1999), feedbacks with the terrestrial biosphere (Cox et al. 2000) and weakening carbon sinks at higher warming levels (Fung et al. 2005)."

*Furthermore, Silvy et al.*
*https://egusphere.copernicus.org/preprints/2024/egusphere-2024-488/, which includes*
*several authors from the present paper, recently published a model intercomparison paper,*
*AERA-MIP, involving participation from over 10 ESMs conducting emission-driven*
*simulations. This contribution deserves acknowledgment.*

Good point - we add a section on adaptive approaches:

**4.3.1    Adaptive approaches**

The discussion throughout this study has focused on prescribed scenarios, both idealised and quasi-realistic as generated by Integrated Assessment Models in the CMIP ScenarioMIP exercise.  In this model, the ensemble of Earth System Models acts as a measure of uncertainty in the coupled carbon-climate response to the emissions pathway.  However, the emissions-driven approach opens the door to more interactive treatment of emissions reduction as a function of realised climate change.  Since the Paris Agreement, some literature has focussed more on adaptive approaches which allow for convergence of a climate model to a target.  Such approaches have been used extensively in simple climate models where it is computationally easy to solve for a given target (Sanderson, O'Neill, and Tebaldi 2016; Avrutin, Goodwin, and Ezard 2023), and in hybrid mode where simple climate models tuned to reproduce the coupled dynamics of an ESM are used to produce custom emissions pathways for an Earth System Model which are consistent with a given temperature target (Sanderson et al. 2017).

A recent proposal "AERA-MIP" (Silvy et al. 2024) has proposed an interactive adaptive approach, where emissions are adjusted in an Earth System Model simulation using the relationship between cumulative emissions and temperature (Matthews et al. 2009) to interactively compute an emissions trajectory consistent with the remaining carbon budget.  Though this approach is a simplistic model for mitigation policy response to experienced climate change, it opens the door for more complex adaptive policy scenarios in the future in which there exist two way couplings between the societal/technological representation.  Such adaptive approaches are increasingly under consideration in the IAM literature (Gambhir et al. 2023) and some groups have succeeded in partial coupling of an ESM and IAM in an integrated framework.(W. D. Collins et al. 2015). Future efforts could explore more fully the interactions between experienced climate impacts, mitigation ambition and capacity.

1. *Benefit of having CO2-emission driven runs*

*I found it somewhat surprising that Section 2.1 does not touch upon some rather*
*fundamental implications of emissions-driven runs. For instance, these simulations enable a*

*more robust assessment of uncertainties in future projections of ocean acidification. In the IPCC AR6 WG I report, global surface pH projections lacked uncertainties due to the prescription of atmospheric CO2. However, with emissions-driven runs, model uncertainty can now be quantified. This holds for various Earth system processes dependent on atmospheric CO2 concentration.*

Agreed this is an omission - added the following:

**The IPCC AR6 WG I report highlighted the limitations of concentration-driven experiments in CMIP6 for projecting future ocean acidification(Intergovernmental Panel on Climate Change (IPCC) 2023). Inter-model variance in surface pH is very low in a given scenario (Lovenduski et al. 2016), largely because all ocean models experience identical surface CO2 concentrations (Kwiatkowski et al. 2020). Emissions-driven simulations would represent the full joint dynamics of ocean and atmosphere heat and carbon evolution. Such factors would represent an improvement in the categorisation of uncertainty in any Earth System processes which are directly or indirectly dependant on atmospheric CO2 concentrations.**

1. *Section 'A proposal for an ESM-DECK'*

*I found the proposal for the new scenarios lacks a bit of novelty and inspiration. It surprised me that innovative adaptive approaches, developed in recent years and potentially offering more policy-relevant scenarios and simulations, were not considered (see for example Silvy et al. 2024 or Terhaar et al. 2022, NCC). Although it is stated on lines 594-596 that these idealized emission-driven experiments would facilitate the calculation of key carbon-climate metrics for informing climate policy tools like the IPCC remaining carbon budget for climate stabilization, the proposed simulations do not stabilize at predefined global warming levels. This omission makes it challenging to calculate the remaining carbon budget and associated metrics essential for climate policy. Adaptive approaches, in contrast, appear more appealing and novel. I was wondering, why this has not been addressed.*

We've added a discussion of adaptive approaches in the 'CMIP8 and beyond section'. We've reduced the ESM DECK section a lot in this version (to be included in a dedicated flat10 paper), but the ESM-DECK section is specifically about diagnostic experiments for the assessment of TCRE, ZEC etc.

*Moreover, a fixed CO2 emission rate of 10 GtC/yr seems excessively high, far exceeding the average historical levels. I am curious about the rationale behind such a high emission rate.*

The emissions rate matches approximately current anthropogenic emissions - translating to the current observed warming rate with the correct value of TCRE. This is now clarified in the text.

*As the main authors themselves demonstrate, zero emissions commitment (ZEC) likely manifests during the emission ramp-down. However, if I understand correctly, ZEC will be diagnosed from simulations branching off from the flat10 experiment. This raises questions about the policy relevance of ZEC in this context.*

The primary goal of the three experiments is to provide clean idealised metrics, which together provide diagnostics of the transient, cessation and reversibility properties of the model. But - we agree that a policy-relevant experiment would consider an idealised gradual approach to net zero. As such, we propose an additional experiment:

"A final experiment, *esm-flat10-nz,* would branch in year 150 from esm-flat10-cdr, allowing an assessment of zero-emissions response under an idealized gradual decline from current emissions rates to net zero. This experiment would provide a companion experiment to *esm-flat10-zec,* assessing how zero emissions response differs between an instantaneous cessation and a gradual approach to net zero (Koven, Sanderson, and Swann 2023)."

*In addition, Joos et al. (2013) have shown that a pulse of 100 Pg C is relatively small, with the temperature response still containing significant natural variability compared to the signal. Wouldn't it be more beneficial to use a larger pulse?*

We're no longer recommending a Joos pulse experiment, but we include the CMIP6 100PgC experiment in the plot for context.

*Furthermore, emission-driven runs are also planned for TIPMIP ESM simulations, aiming for a fixed warming rate rather than a fixed emission rate. What are the similarities and dissimilarities between these two approaches?*

We are in close consultation with TIPMIP to ensure there's maximum utility with these two experiments. Fundamentally, the TIPMIP approach is well suited to temperature framing (climate changes observed at prescribed warming rates, and for the assessment of tipping points and climate impacts at defined warming levels). The design allows for a warming level to be reached in a predictable time, with comparable rates of ocean heat accumulation in the participating climate models.

 The flat10 approach is suited to questions of carbon framing - all models are subject to the same boundary conditions, which means that TCRE and ZEC can be cleanly assessed without interactive experiments or model-specific cumulative emissions. Questions of carbon system coupling, fraction of carbon accumulating in different system pools are cleanly estimated. Flat10 is also a much smaller request than TIPMIP, specifically designed to return a small set of carbon-climate feedback metrics.

*L.44: 'represented upstream of ESMs' Unclear what upstream of ESMs means. Try to clarify.*

Rewritten:

  Carbon removal and sequestration strategies, which rely on proposed human management of natural systems, are currently calculated in IAMs during scenario development with only the net carbon emissions passed to the ESM.

*L 60: Reference should be fixed; i.e. n.d.*

Will address

*L 75-76: Reference should be fixed; i.e. n.d.*

Will address

*Figure 1: This figure seems to be inspired by Figure 1 of a paper submitted in September 2023 by Pfleiderer et al. (the main author and some co-authors are also co-authors on the Pfleiderer paper). This needs to be acknowledged.*

Done

*L447: This is not true for AERA-MIP.*

Added Silvy reference here.

*L667: I know that this argument has already been made during the ZECMIP discussion. I still do not understand this. Are the technical challenges substantial (which I doubt)?*

Perhaps not substantial, but definitely requiring more human interaction time to automate the experiments, both in simple climate models and in ESMs.

---

## Author Response (AR2)

**Response to minor Revisions:**

**Response to Malte Meinshausen:**

Thanks and congrats for a very nice manuscript again. I have nothing more to add given that the authors addressed all the earlier comments.

Just some minor point I noticed that you want to address (can be during proof-reading). In your revised Figure 5, the middle row of panels has the wrong y-axis labels, as the normalisation period is 1850-1900, not 1970-1990. And the figure panels letters should be 'b' and 'c'.

*Many thanks for your help with the manuscript - the issues with the Figure have been corrected.*

**Response to Reviewer #2**

*Many thanks again for the reviewer's time in improving our manuscript. The remaining minor issues are addressed as outlined below:*

L 54: what about the nitrogen cycle? I suggest adding it here too.
*Done*

L60: maybe say km-scale climate models
*Done*

L72-74: This sentence is difficult to read. Please rewrite.
*Done as follows:*
*" In ScenarioMIP/CMIP6, scenarios were defined  in terms of SSPs representing broad socioeconomic background states combined with global mean end-of-century radiative forcing targets (O'Neill et al., 2016; Riahi et al., 2017). "*

L 188ff: This section discusses RCPs, whereas the previous section focused mostly on SSPs. Then, on line 191, it switches back to SSPs. I found this confusing.

*Agreed. We've cut the offending line.*
L193: 'excludes carbon cycle uncertainties'. This is not entirely accurate. The changes in land and ocean carbon stocks still show a large spread in responses, even under concentration-driven simulations. However, these uncertainties do not feedback on atmospheric CO2 and climate. I believe this needs to be clarified.

*Revised as follows:*
*"This has pragmatic advantages in terms of coordinating research across climate disciplines, but excludes uncertainties arising from feedbacks from the carbon cycle back onto atmospheric $CO_2$. "*

L207: Clarify that this sentence is about CO2 emissions and not any other emissions.
*Specified 'carbon' emissions*

L210-212: Not sure if I follow the argument here. If they are idealized simulations, why should they apply to real-world carbon-cycle dynamics?

*Agreed - removing this sentence (which referred more to the ESM DECK proposal, less relevant to the scenario focus of the final paper)*

L214-216: Can you quantify this?

*Changed figure A1 to illustrate this better with post-2014 cumulative compatible emissions.*

L222-224: I am confused here. Why would the model output on the biome level be necessary to resolve the complete carbon budget? I guess output on the biome level (e.g. ocean or land biomes) would only be useful if you would like to calculate a regional ocean/land carbon budget. But for this, you would probably also need regional ocean interior carbon changes. Maybe I miss here something.

*Rewritten as follows: "However, CMIP6 models remain inconsistent in their outputting, unit conventions and definitions of component-level carbon fluxes, which complicate analysis. Such issues must be better addressed in emissions-driven simulations where reconstruction of the carbon budget is of first order importance to understanding the model response"*

L. 257: I guess the carbon-concentration and carbon-climate feedback parameter uncertainty was obtained from Arora et al. 2020, rather than Jones et al. or Friedlingstein 2006. Is this assumption correct?

*Agreed, corrected*

L278: Maybe add here references to Allen et al. 2009 and Matthews et al. 2009, which introduced TCRE.

*Done*

L761: Maybe add here reference to Terhaar et al. (2023, ERL; https://iopscience.iop.org/article/10.1088/1748-9326/acaf91)

*Done, thanks.*

L1015-1016: Please mention the technical challenges explicitly or delete this point (1).

*Deleted the reference to technical challenge*

L1015-1020: These are all very technical points to justify the esm-flat10 experiments, but they are easily solvable. I think point (4) is the most convincing argument and should be listed first.
*Fair enough - changed as suggested.*

L1023-1027: Maybe I am missing something here. But using the 1%CO2 run to calculate TCRE and ZEC as well as AF was 'a consistent set of experiments' . So I am struggling to understand why the esm-flat10 esm-flat10-zec experiments would be more consistent.

*Deleting this paragraph*

Figure 7: I think the y-axis labels for the middle row panels are wrong. Should be vs 1850-1900

*Fixed*

Figure 7: I am a bit surprised about the behavior of the CNRM-ESM2-1 model, which seems to be quite an outlier and simulates an atmospheric CO2 concentration of only about 340 ppm in 2010. What are the reasons for this? Why does the model still simulate a temperature change over the historical period, which is about consistent with observations?

*Beyond scope to diagnose the reasons here - but we've double checked the numbers.*

L1098. Yes, 10 out of 13 models. However, the 3 models that are not included here show very strong biases. Would you argue, that such models should be excluded in any subsequent analysis?

*Yes - added the following:*
*"we suggest that the remaining outlier models may require greater attention to calibration of historical CO2 concentrations if emissions-driven simulations are the only runs provide and we suggest that the remaining outlier models may require greater attention to calibration of historical CO2 concentrations if emissions-driven simulations are the only runs provided."*

L1135: N2O is mentioned here, but not in the abstract.

*fixed*
L1693: Add Goodwin et al. 2018 (https://agupubs.onlinelibrary.wiley.com/doi/full/10.1002/2017EF000732) and Terhaar et al. (2022; https://www.nature.com/articles/s41558-022-01537-9)

*Done, thanks.*

L 1697: rather write 'consistent with any prescribed global warming target'

*Agreed, changed as suggested.*

L 1697-1700: It also allows for emission-driven simulations that can stabilize temperature at various global warming levels, enabling the assessment of impacts at different degrees of warming.

*Added - thanks.*

---

## Author Response (AR3)

Dear editor,

Apologies for the delay - technical correction to provide a current zenodo link to the code for the study is now included.